New insights into the cranial osteology of the Early Cretaceous paracryptodiran turtle Lakotemys australodakotensis

Rollot Yann yann.rollot@gmail.com 1
Evers Serjoscha W. 1
Cifelli Richard L. 2
Joyce Walter G. 1
1 Department of Geosciences, University of Fribourg , Fribourg , Switzerland
2 Oklahoma Museum of Natural History , Norman , OK , USA
Abdala Virginia
Electronic publication date: 2022 Apr 12
Publication date: 2022
Volume: 10
Electronic Location ID: e13230
Received 2021 Nov 19; Accepted 2022 Mar 16
Copyright: ©2022 Rollot et al.
Copyright year: 2022
Copyright holder: Rollot et al.
License: This is an open access article distributed under the terms of the Creative Commons Attribution License, which permits unrestricted use, distribution, reproduction and adaptation in any medium and for any purpose provided that it is properly attributed. For attribution, the original author(s), title, publication source (PeerJ) and either DOI or URL of the article must be cited.
License URL: https://creativecommons.org/licenses/by/4.0/

Keywords: Testudinata, Paracryptodira, Baenidae, Anatomy, Systematics, Turtles

Funding: Swiss National Science Foundation SNF 200021_178780/1 American Chemical Society Petroleum Research Fund ACS–PRF#38572–AC8 U.S. National Science Foundation FRES–1925896 Yann Rollot, Serjoscha W. Evers, and Walter G. Joyce were supported by the Swiss National Science Foundation (SNF 200021_178780/1). Richard L. Cifelli was supported by the American Chemical Society Petroleum Research Fund (ACS–PRF#38572–AC8) and the U.S. National Science Foundation (FRES–1925896). The funders had no role in study design, data collection and analysis, decision to publish, or preparation of the manuscript.

==============================
Lakotemys australodakotensis is an Early Cretaceous paracryptodire known from two shells and a skull from the Lakota Formation of South Dakota, USA. Along with the Early Cretaceous Arundelemys dardeni and the poorly known Trinitichelys hiatti, Lakotemys australodakotensis is generally retrieved as an early branching baenid, but more insights into the cranial anatomy of these taxa is needed to obtain a better understanding of paracryptodiran diversity and evolution. Here, we describe the skull of Lakotemys australodakotensis using micro-computed tomography to provide the anatomical basis for future phylogenetic analyses that will be needed to investigate more precisely the intrarelationships of Paracryptodira. Preliminary comparisons reveal that the cranial anatomy of Lakotemys australodakotensis is very similar to that of the Aptian-Albian basal baenid Arundelemys dardeni, that both taxa exhibit a remarkable combination of derived characters found in baenodds and characters found in non-baenid paracryptodires, particularly Pleurosternidae, and that Lakotemys australodakotensis is the only known baenid to date to possess a canal for the palatine artery.

Introduction

Baenidae Cope, 1873 is a clade of freshwater turtles that lived in North America from the Early Cretaceous to the Eocene and currently comprises about 30 valid species (Hay, 1908; Gaffney, 1972a; Gaffney, 1972b; Joyce & Lyson, 2015). Baenid turtles were particularly diverse from the Campanian to the Eocene, but the fossil record of its basal representatives in the Early Cretaceous remains scarce. At present, only four baenid taxa are known from the Early Cretaceous of North America: Protobaena wyomingensis (Gilmore, 1920) from the Albian of Wyoming, Arundelemys dardeni (Lipka et al., 2006) from the Aptian-Albian of Maryland, Trinitichelys hiatti Gaffney, 1972a from the Aptian-Albian of Texas, and Lakotemys australodakotensis Joyce, Rollot & Cifelli, 2020 from the Berriasian-Valanginian of South Dakota. Lakotemys australodakotensis was erected based on two partial shells (OMNH 67133, the holotype, and OMNH 63615) and a poorly preserved skull (OMNH 66106). Although most sutures were easily identified in the shells with the use of X-ray micro-computed tomography (µCT), the interpretation of the internal suture interfaces of the skull was obstructed by a thick layer of metal oxides, which produce shiny white spots in the coronal slices, occasionally poor preservation of the internal structures of the bones, and also partial crushing and shearing of the skull itself. These aspects lead Joyce, Rollot & Cifelli (2020) to provide only a brief description of OMNH 66106 that pertains to the general size and shape of the skull, orbits, and temporal emarginations, and the identification of portions of the frontal, postorbital, parietal, and squamosal bones. The cranial anatomy of basal baenids in general remains poorly known to date, with the exception of Arundelemys dardeni, which recently benefited from two descriptive studies (Lipka et al., 2006; Evers, Rollot & Joyce, 2021). In contrast, the cranial anatomy of more advanced baenids from the Campanian and Maastrichtian of North America, especially Alberta, New Mexico, Utah, Wyoming, Montana, North Dakota, and South Dakota (Joyce & Lyson, 2015), has received far greater attention over the last three decades (Brinkman, 2003; Brinkman & Nicholls, 1991; Brinkman & Nicholls, 1993; Lively, 2015; Lyson & Joyce, 2009a; Lyson & Joyce, 2009b; Lyson & Joyce, 2010; Lyson, Sayler & Joyce, 2019; Lyson et al., 2016; Lyson et al., 2021; Rollot, Lyson & Joyce, 2018). A notable morphological and temporal gap, which remains to be bridged, exists from the Cenomanian to the Coniacian between advanced baenids and their poorly known basal relatives (Joyce & Lyson, 2015). New insights into the anatomy of basal baenid taxa is, therefore, crucial for better understanding the evolution of baenids in particular, and paracryptodires in general.

As part of a current project that aims to investigate paracryptodire anatomy, systematics, and relationships, we here provide a detailed description of OMNH 66106, the only known skull of Lakotemys australodakotensis. Despite the abovementioned, initial difficulties in interpreting the µCT image stack of OMNH 66106 (Joyce, Rollot & Cifelli, 2020), we are now able to describe almost every cranial bone individually and provide preliminary comparative insights with other known paracryptodiran skulls.

Material and Methods

We used the original set of µCT scans obtained by Joyce, Rollot & Cifelli (2020) for their initial study of Lakotemys australodakotensis. OMNH 66106 was scanned at the University of Texas High-Resolution X-ray Computed Tomography Facility with 1,400 projections over 360°, a voltage of 210 kV, and a current of 170 µA. 1029 coronal slices were obtained with a voxel size of 55 µm for the x and y axes, and 59 µm for the z axis, contra Joyce, Rollot & Cifelli (2020) who only mentioned a voxel size of 59 µm. The specimen was segmented using the software Amira (version 2019.2; https://www.thermofisher.com/us/en/home/electron-microscopy/products/software-em-3d-vis/amira-software.html) and the reconstructions were obtained through every fifth slice segmentation and use of the interpolation tool in the x, y, and z-axes. The 3D models were exported as .ply-files and the software Blender 2.79b (https://www.blender.org) was used to create the images used in the figures. The µCT slice data and 3D models generated as part of this study are available at MorphoSource (https://www.morphosource.org/projects/000379242?).

We compare OMNH 66106 to a selection of putative paracryptodires, including compsemydids and helochelydrids, for which cranial material is known. The taxa, specimens, and references used for comparative purposes are listed in Table 1. Hypothesized paracryptodiran relationships are recapitulated in the phylogenetic trees shown in Fig. 1. A novel hypothesis of paracryptodiran relationships, which includes the observations made here, are planned for a future contribution.

Systematic Palaeontology

TESTUDINATA Klein, 1760 (Joyce et al., 2020)	
PARACRYPTODIRA Gaffney, 1975 (Joyce et al., 2021)	
LAKOTEMYS (Joyce, Rollot & Cifelli, 2020)	
Lakotemys australodakotensis (Joyce, Rollot & Cifelli, 2020)	

Holotype: OMNH 67133, a partial shell (Figs. 2, 4A, Joyce et al., 2020).

Table 1 List of comparative material used in this study.

Abbreviations for the collections housing the specimens are provided in the Institutional Abbreviations section.

Taxon	Specimen(s)	Geologic age	Geographic origin	Citation(s)	
Compsemydidae					
Compsemys victa	UCM 49223	Maastrichtian	Colorado, USA	Lyson & Joyce (2011)	
					
Pleurosternidae					
Dorsetochelys typocardium	DORCM G23	Berriasian	England	Evans & Kemp (1976)	
Glyptops ornatus	AMNH 336, YPM 1784, YPM 4717	Tithonian	Wyoming, USA	Gaffney (1979)	
Pleurosternon bullockii	UMZC T1041	Berriasian	England	Evans & Kemp (1975)	
				Evers, Rollot & Joyce (2020)	
Pleurosternon moncayensis	MPZ 2020/53	Tithonian-Berriasian	Spain	Perez-Garcia et al. (2021)	
Uluops uluops	UCM 53971	Tithonian	Wyoming, USA	Rollot, Evers & Joyce (2021a)	
Helochelydridae					
Helochelydra nopcsai	IWCMS 1998.21	Barremian	England	Joyce et al. (2011)	
Naomichelys speciosa	FMNH PR273	Aptian-Albian	Texas, USA	Joyce, Sterli & Chapman (2014)	
Baenidae					
Arundelemys dardeni	USNM 41614	Aptian-Albian	Maryland, USA	Lipka et al. (2006)	
				Evers, Rollot & Joyce (2021)	
Trinitichelys hiatti	MCZ 4070	Aptian-Albian	Texas, USA	Gaffney (1972a); Gaffney (1972b)	
Neurankylus eximius	UALVP 30824, TMP 89.36.112	Campanian	Alberta, Canada	Brinkman & Nicholls (1993)	
Neurankylus torrejonensis	NMMNH P-9049	Danian	New Mexico, USA	Lyson et al. (2016)	
Arvinachelys goldeni	UMNH VP 21151,	Campanian	Utah, USA	Lively (2015)	
	UMNH VP 21300				
Hayemys latifrons	AMNH 6139	Maastrichtian	Wyoming, USA	Gaffney (1972a); Gaffney (1972b)	
Baena arenosa	MCZ 4072	Lutetian	Wyoming, USA	Gaffney (1972a); Gaffney (1972b)	
Boremys pulchra	TMP 88.2.10, TMP 80.16.1,	Campanian	Alberta, Canada	Brinkman & Nicholls (1991)	
	TMP 79.14.1053				
Chisternon undatum	AMNH 5961	Ypresian-Lutetian	Wyoming, USA	Gaffney (1972a); Gaffney (1972b)	
Eubaena cephalica	DMNH 96004	Maastrichtian	North Dakota, USA	Rollot, Lyson & Joyce (2018)	
Goleremys mckennai	UCMP 179519	Thanetian	California, USA	Hutchison (2004)	
Saxochelys gilberti	DMNH EPV.96000,	Maastrichtian	North Dakota, USA	Lyson, Sayler & Joyce (2019)	
	DMNH EPV. 130939,				
	DMNH EPV.97041				
Stygiochelys estesi	AMNH 2601	Maastrichtian	Montana, USA	Gaffney (1972a); Gaffney (1972b)	
Palatobaena bairdi	CCM 77-11	Selandian	Montana, USA	Archibald & Hutchison (1979)	
Palatobaena cohen	YPM 57498, DMNH EPV.96002 DMNH EPV.97016, DMNH EPV.97017	Maastrichtian	North Dakota, USA	Lyson & Joyce (2009a)	
Palatobaena gaffneyi	UCMP 114529	Ypresian	Wyoming, USA	Archibald & Hutchison (1979)	
Palatobaena knellerorum	DMNH EPV.134081	Danian	Colorado, USA	Lyson et al. (2021)	
Cedrobaena brinkman	UMMP 20490, DMNH EPV.96003	Maastrichtian	Montana & North Dakota, USA	Lyson & Joyce (2009b)	
Cedrobaena putorius	FMNH PR2258, DMNH EPV.97018	Maastrichtian	North Dakota & South Dakota, USA	Lyson & Joyce (2009b)	
Gamerabaena sonsalla	ND06-14.1	Maastrichtian	North Dakota, USA	Lyson & Joyce (2010)	
Plesiobaena antiqua	TMP 99.55.145, TMP 81.41.103	Campanian	Alberta, Canada	Brinkman (2003)	

Type locality and horizon: OMNH locality V1332, Dick Canyon, Fall River County, South Dakota, USA; Unit 2, Chilson Member, Lakota Formation, Berriasian–Valanginian. See Joyce, Rollot & Cifelli (2020) for additional information.

Revised diagnosis: Lakotemys australodakotensis can be diagnosed as a representative of Paracryptodira by the presence of a finely and densely sculptured skull and shell and the location of the entrance foramina of the carotid artery branches into the skull about mid-length along the pterygoid-parabasisphenoid suture, and as a representative of Baenidae by the absence of epiplastral processes, the development of well-developed axillary and inguinal buttresses, the presence of a posterior process of the pterygoid with an elongate contact with the basioccipital, the absence of secondary basioccipital tubera formed by the parabasisphenoid, and the absence of a basipterygoid process. Among named Early Cretaceous baenids, Lakotemys australodakotensis can be differentiated from Protobaena wyomingensis and Trinitichelys hiatti by having an irregularly shaped vertebral V that does not lap onto neural VIII and that forms two anterolateral processes that partially hinder vertebral IV from contacting pleural IV; and can be differentiated from Arundelemys dardeni and Trinitichelys hiatti by having a slightly broader skull, a very shallow pterygoid fossa, and a canal for the palatine artery. A feature that is distinct from all known paracryptodires is a trigeminal foramen that is completely enclosed by the pterygoid and prootic medially, but the pterygoid and parietal laterally.

Figure 1 Phylogenetic hypotheses of paracryptodiran relationships.

(A) Simplified strict consensus tree resulting from the analyses of Perez-Garcia, Royo-Torres & Cobos (2015: Fig. 5D). (B) Simplified single most parsimonious tree resulting from the analyses of Rollot, Evers & Joyce (2021a): 10B). Lakotemys australodakotensis was manually grafted on both trees following results of Joyce, Rollot & Cifelli (2020). The range of North American taxa is highlighted in blue, the range of European taxa in red, and the range of clades that have both a European and North American distribution in black.

Figure 2 OMNH 66106, skull of Lakotemys australodakotensis, Early Cretaceous (Berriasian-Barremian) of South Dakota, USA.

Three-dimensional renderings of the skull embedded in matrix and with matrix rendered transparent in (A) dorsal view, (B) ventral view, (C) left lateral view, (D) anterior view, and (E) posterior view.

Referred material: OMNH 66106, a preserved skull collected from the type locality; OMNH 63615, a poorly preserved shell collected from OMNH site V1382. All referred material originates from Unit 2 of the Chilson Member of the Lakota Formation (Joyce, Rollot & Cifelli, 2020).

Description

General comments

The skull of OMNH 66106 is embedded in a dense sandstone matrix but remains nearly complete (Fig. 2). Most of the preserved bones remain in articulation despite some shearing and dorsoventral compression. The anteroventral part of the skull is damaged. In particular, the anterior portions of both maxillae are missing, as are the anterior and medial portions of the left palatine, and the ventral portions of the prefrontals (Figs. 2C and 2D). The mandibular condyles have eroded and the vomer is missing completely. With the exception of the left prootic, opisthotic and quadrate, as well as the basioccipital and exoccipitals, which were respectively segmented as one block, all the other bones of OMNH 66106 can be observed individually (Fig. 3). The surface of most skull roof bones appears to be slightly eroded, which hinders any clear observation of the skull ornamentation. The shearing that affects the skull does not allow reconstruction of the original shape of the labyrinth, but most of its anatomical structures can nevertheless be described properly. Despite difficulties in interpreting the µCT scans, the image stack and 3D reconstructions were checked by three of the authors and, unless mentioned otherwise below in some particular cases, we are confident about the reconstructions and interpretations provided herein. We note that the right side of OMNH 66106 is overall better preserved than the left side, and therefore base our observations and descriptions on the former when different reconstructions are apparent on each side. A list of basic measurements is provided in Table 2. These must be viewed with caution as OMNH 66106 shows much shearing and compression.

Figure 3 OMNH 66106, skull of Lakotemys australodakotensis, Early Cretaceous (Berriasian-Barremian) of South Dakota, USA.

Bone by bone three-dimensional renderings of the skull in (A) dorsal view, (B) ventral view, (C) left lateral view, (D) right lateral view, (E) anterior view, and (F) posterior view. Abbreviations: bo-ex, basioccipital-exoccipital complex; epi, epipterygoid; fr, frontal; fpp, foramen palatinum posterius; fst, foramen stapedio-temporale; ica, incisura columella auris; ju, jugal; mx, maxilla; na, nasal; op, opisthotic; pa, parietal; pal, palatine; pbs, parabasisphenoid; pf, prefrontal; po, postorbital; poq, prootic-opisthotic-quadrate; pro, prootic; pt, pterygoid; qj, quadratojugal; so, supraoccipital; sq, squamosal.

Figure 4 Surface patterns of OMNH 66106.

(A) Dorsal view of the segmented skull showing the location of the area of interest highlighted in B. (B) Dorsal view of the posterior portion of the right parietal. (C) Left lateral view of the segmented skull showing the location of the area of interest highlighted in D. (D) Lateral view of the left quadratojugal. The surface patterns are highlighted by the black arrows.

The skull of OMNH 66106 is longer than wide (Figs. 2A, 2B and 3A) and, given its proportions, resembles that of Trinitichelys hiatti (Gaffney, 1972a), Arundelemys dardeni (Lipka et al., 2006), and Pleurosternon bullockii (Evans & Kemp, 1975), and differs from the more elongate skull of Glyptops ornatus (Gaffney, 1979) and the wedge-shaped skull of baenodds (Joyce & Lyson, 2015) and Uluops uluops (Rollot, Evers & Joyce, 2021a). The left quadratojugal and posterior portion of the right parietal seem to exhibit some surface pattern, but the following observations have to be taken with caution because of the overall bad preservation of the skull bone surfaces. The apparent pattern along the posterior margin of the right parietal consists of some kind of striation (Figs. 4A and 4B), which is reminiscent of the ridge-like pattern in the same area in Pleurosternon bullockii (Evers, Rollot & Joyce, 2020), while the overall surface texture of the left quadratojugal (Figs. 4C and 4D) resembles that on the same bone in Uluops uluops (Rollot, Evers & Joyce, 2021a). Paracryptodires are known to exhibit various surface textures, from irregular tubercles in Glyptops ornatus (Gaffney, 1979), Pleurosternon bullockii (Evers, Rollot & Joyce, 2020), Pleurosternon moncayensis (Perez-Garcia et al., 2021) and Uluops uluops (Rollot, Evers & Joyce, 2021a), low and irregular pits in Arundelemys dardeni (Evers, Rollot & Joyce, 2021), to a smooth texture in the majority of derived taxa (Joyce & Lyson, 2015), but poor preservation in OMNH 66106 does not allow us to compare its ornamentation pattern to any of the latter with confidence. The orbits were likely once oriented laterally when taking dorsoventral compression of the fossil into consideration (Figs. 2B and 3A). As such, the orientation of the orbits resembles that of Arundelemys dardeni (Lipka et al., 2006; Evers, Rollot & Joyce, 2021), Compsemys victa (Lyson & Joyce, 2011), Uluops uluops (Rollot, Evers & Joyce, 2021a), and Saxochelys gilberti (Lyson, Sayler & Joyce, 2019), but differs from that of Eubaena cephalica (Rollot, Lyson & Joyce, 2018), Palatobaena cohen (Lyson & Joyce, 2009a), Palatobaena knellerorum (Lyson et al., 2021), and Cedrobaena putorius (Lyson & Joyce, 2009b). The upper temporal emargination is deep, with the foramen stapedio-temporale being exposed in dorsal view, and the emargination slightly extending beyond the anterior margin of the cavum tympani (Figs. 2B and 3A). A deep upper temporal emargination is also present in all baenodds but Baena arenosa (Gaffney, 1972a; Hutchison, 2004; Joyce & Lyson, 2015; Lyson, Sayler & Joyce, 2019; Lyson et al., 2021) and contrasts with the condition observed in Dorsetochelys typocardium (Evans & Kemp, 1976), Neurankylus torrejonensis (Lyson et al., 2016), Pleurosternon bullockii (Evans & Kemp, 1975), and Uluops uluops (Rollot, Evers & Joyce, 2021a). The cheek emargination of OMNH 66106 is relatively deep, just reaching the lower margin of the orbit (Figs. 3C and 3D), as observed in most baenodds (Archibald & Hutchison, 1979; Brinkman & Nicholls, 1991; Brinkman, 2003; Gaffney, 1972a; Hutchison, 2004; Lively, 2015; Lyson & Joyce, 2009a; Lyson et al., 2021; Rollot, Lyson & Joyce, 2018) with the exception of Baena arenosa and Chisternon undatum, in which the cheek emargination is deeper than in OMNH 66106. Among non-baenid paracryptodires, a deeper cheek emargination is found in Pleurosternon bullockii (Evans & Kemp, 1975) and Uluops uluops (Carpenter & Bakker, 1990) whereas Compsemys victa lacks any cheek emargination (Lyson & Joyce, 2011).

Table 2 List of measurements made on OMNH 66106.

	Measurements (mm)	
Skull length (nasal to supraoccipital)	56	
Skull width at quadrates	48	
Skull height at quadrates	22	
Foramen magnum width	6	
Foramen magnum height	6	

Figure 5 Three-dimensional renderings of isolated bones from the anterior skull roof area of OMNH 66106.

(A) Dorsolateral view of the right frontal, prefrontal, nasal, and maxilla. (B) Dorsolateral view of the left frontal, prefrontal, nasal, and maxilla. (C) Ventral view of the frontals and right prefrontal. (D) Anterior view of the frontals and left prefrontal. (E) Anterior view of the nasals, frontals, and prefrontals. Abbreviations: cc, crista cranii; dppf, descending process of the prefrontal; fr, frontal; mx, maxilla; na, nasal; pf, prefrontal; sol, sulcus olfactorius.

Nasal

The nasal is a small, triangular element that forms the dorsal margin of the external naris and roofs the fossa nasalis anteriorly (Figs. 3A, 3B and 3E). The nasal is as long as wide and resembles that of other non-baenodd paracryptodires. In dorsal view, the nasal contacts its counterpart medially, the anterior process of the frontal posteromedially, and the dorsal plate of the prefrontal posterolaterally (Fig. 3A). The nasal contacts the ascending process of the maxilla laterally on the right side (Fig. 5A), but such a contact is prevented on the left side by a thin anterior process of the prefrontal that reaches the margin of the external naris (Figs. 3C–3C and 5B). As the nasal-maxilla contact is only absent on the left side, and given that this area is particularly difficult to segment, it is likely that only one of the two observed conditions is correct. We are not able to favor one side reconstruction over the other as our segmented models correspond to what is visible in the µCT slice data, but note that the absence of a nasal-maxilla contact is highly unusual as all known paracryptodires with distinct nasals exhibit a nasal-maxilla contact along the anteriormost aspect of the skull (Gaffney, 1972a; Evans & Kemp, 1975; Evans & Kemp, 1976; Lyson & Joyce, 2011; Joyce & Lyson, 2015; Rollot, Evers & Joyce, 2021a), with the exception of Naomichelys speciosa and Helochelydra nopcsai, where the prefrontal prevents the two bones from contacting each other (Joyce et al., 2011; Joyce, Sterli & Chapman, 2014). This contact may be absent in numerous baenodds as well (e.g., Chisternon undatum, Gaffney, 1972a), but is difficult to assess, given that the nasal is often fused to the frontal (Gaffney, 1972a). The anterior process of the frontals moderately protrudes between the nasals posteriorly, separating the nasals for half of their length (Fig. 3A), which is reminiscent of the condition observed in Arundelemys dardeni (Lipka et al., 2006; Evers, Rollot & Joyce, 2021), Eubaena cephalica (Rollot, Lyson & Joyce, 2018), Neurankylus torrejonensis (Lyson et al., 2016), and Pleurosternon bullockii (Evers, Rollot & Joyce, 2020), but differs from Compsemys victa (Lyson & Joyce, 2011), Dorsetochelys typocardium (Evans & Kemp, 1976), Glyptops ornatus (Gaffney, 1979), Uluops uluops (Rollot, Evers & Joyce, 2021a), and most baenodds (Joyce & Lyson, 2015). Within the nasal cavity, the nasals form along their medial aspect the anterior part of a protruding ridge, otherwise formed by the prefrontals, that divides the cavity into left and right halves.

Prefrontal

The prefrontal forms the anterodorsal margin of the orbit and might have a minor contribution to the dorsolateral margin of the apertura narium externa (see Nasal above; Figs. 3A–3D). The dorsal plate of the prefrontal is exposed on the skull roof as in Arundelemys dardeni (Lipka et al., 2006; Evers, Rollot & Joyce, 2021), Compsemys victa (Lyson & Joyce, 2011), Dorsetochelys typocardium (Evans & Kemp, 1976), Neurankylus torrejonensis (Lyson et al., 2016), Pleurosternon bullockii (Evans & Kemp, 1975; Evers, Rollot & Joyce, 2020), and Uluops uluops (Rollot, Evers & Joyce, 2021a), which contrasts with the reduced to absent exposure of that bone in baenodds (Hutchison, 2004; Joyce & Lyson, 2015; Lyson, Sayler & Joyce, 2019; Lyson et al., 2021). The dorsal plate of the prefrontal contacts the anterior process of the frontal medially and posteriorly, the nasal anteromedially, and the ascending process of the maxilla anterolaterally (Figs. 3A–3D). The descending process of the prefrontal is broken on the left side but seems relatively complete on the right (Figs. 5C–5E). Despite some displacement and damage that affects this area, the descending process likely contacted the palatine ventrolaterally and the vomer ventromedially. Laterally, the descending process of the prefrontal contacts the ascending process of the maxilla. The preserved ventral portion of the descending process is mediolaterally expanded, showing that it formed a thin sheet of bone within the fossa nasalis (Figs. 3E and 5E), as in Arundelemys dardeni (Evers, Rollot & Joyce, 2021) and Uluops uluops (Rollot, Evers & Joyce, 2021a).

Frontal

The frontal is about twice as wide posteriorly as it is anteriorly and forms an anterior process that represents about half of the frontal length (Fig. 3A). The anteriormost third of this process becomes narrower anteriorly and divides the internasal contact for half its length (Figs. 3A and 3B), as observed in Arundelemys dardeni (Lipka et al., 2006; Evers, Rollot & Joyce, 2021), Dorsetochelys typocardium (Evans & Kemp, 1976), and Pleurosternon bullockii (Evans & Kemp, 1975; Evers, Rollot & Joyce, 2020), although the anterior protrusion of the frontal in the latter taxon is slightly deeper. The shape of the anterior process of the frontal is similar to that of Dorsetochelys typocardium (Evans & Kemp, 1976), Arundelemys dardeni (Lipka et al., 2006; Evers, Rollot & Joyce, 2021), and Uluops uluops (Rollot, Evers & Joyce, 2021a), but differs from that of Pleurosternon bullockii, in which the anterior process progressively becomes narrower anteriorly along its full length (Evans & Kemp, 1975; Evers, Rollot & Joyce, 2020). The anterior process contacts its counterpart medially and the dorsal plate of the prefrontal laterally (Figs. 3A, 3B and 5C). The posterior half of the frontal is enlarged for all its length, as in Glyptops ornatus (Gaffney, 1979), and does not form a distinct laterally oriented process that inserts between the prefrontal and postorbital to contribute to the orbit as is the case in Arundelemys dardeni (Lipka et al., 2006; Evers, Rollot & Joyce, 2021), Dorsetochelys typocardium (Evans & Kemp, 1976), Pleurosternon bullockii (Evans & Kemp, 1975; Evers, Rollot & Joyce, 2020), and Uluops uluops (Rollot, Evers & Joyce, 2021a). The contribution of the frontal to the orbit margin in OMNH 66106 is thus slightly greater than that of the aforementioned taxa (Fig. 3A). The posterior half of the frontal contacts the postorbital posterolaterally and the parietal posteriorly (Fig. 3A). The frontal bears a ridge ventrally, the crista cranii (Figs. 3B, 5C and 5D), which is as well-developed as in Arundelemys dardeni (Evers, Rollot & Joyce, 2021), Pleurosternon bullockii (Evers, Rollot & Joyce, 2020), and Uluops uluops (Rollot, Evers & Joyce, 2021a). The crista cranii extends anteroposteriorly within the fossa nasalis and anteriorly joins the mediolateral sheet of bone formed by the descending process of the prefrontal (Fig. 5C). The crista cranii levels off posteriorly and does not form a continuous ridge with the descending process of the parietal, as also observed in Arundelemys dardeni (Evers, Rollot & Joyce, 2021), Pleurosternon bullockii (Evers, Rollot & Joyce, 2020), and Uluops uluops (Rollot, Evers & Joyce, 2021a). The crista cranii of the frontal forms along with its counterpart a deep sulcus olfactorius (Figs. 5C–5E).

Parietal

The parietal forms most of the posterior part of the skull roof and is about twice as long as wide (Fig. 3A). The parietal roofs the braincase dorsally, covers the crista supraoccipitalis for about half its length, and forms most of the deep upper temporal emargination (Fig. 3A). The dorsal plate of the parietal contacts its counterpart medially, the frontal anteriorly, the postorbital laterally, and the crista supraoccipitalis posteromedially (Fig. 3A). A minor contact with the squamosal is apparent on the right side of the skull and a large one was perhaps present on the left (Fig. 3A). A parietal-squamosal contact is also found in Dorsetochelys typocardium (Evans & Kemp, 1976), Pleurosternon bullockii (Evans & Kemp, 1975; Evers, Rollot & Joyce, 2020), Uluops uluops (Rollot, Evers & Joyce, 2021a), Helochelydra nopcsai (Joyce et al., 2011), Naomichelys speciosa (Joyce, Sterli & Chapman, 2014), Neurankylus torrejonensis (Lyson et al., 2016), and the baenodds Baena arenosa and Chisternon undatum (Gaffney, 1972a). Although a parietal-squamosal contact is present in OMNH 66106, the posterior margin of the parietal is deeply excavated, which exposes the foramen stapedio-temporale in dorsal view (Fig. 3A).

Our reconstructions of the parietal and postorbital highlight some differences along the sutural aspect between the two bones on the left and right sides. Whereas the parietal-postorbital external suture seam is continuously curved on the right side in dorsal view, it is strongly irregular on the left, and the posteriormost aspect of the left parietal is also slightly wider than that of its right counterpart (Fig. 3A). As it is unusual for the postorbital to have strong medial deflections into the parietal, we consider the morphology on the right side to reflect the anatomical arrangement in life more accurately. Nevertheless, we segmented the models along the visible sutural contacts in the CT scans, and thus leave these models asymmetrical to highlight uncertainties in the reconstruction. Overall, these uncertainties are minor and do not affect important interpretations, such as the contact with the squamosal, which is unambiguously present on both sides. Within the braincase, the “rider,” a protuberance located dorsal and anterior to the cerebellar part of the brain-endocast, is well-demarked as a triangular depression (morphotype IV of Werneburg, Evers & Ferreira, 2021), as in Plesiobaena antiqua (Gaffney, 1982), but different from the elongated “rider” one of Eubaena cephalica (Werneburg, Evers & Ferreira, 2021).

The descending process of the parietal forms the posterior margin of the foramen interorbitale, the anterior part of the lateral wall of the braincase, and the medial margin of the fossa temporalis. Slightly posterior to the posterior margin of the foramen interorbitale, the parietal bears a prominent ridge on its lateral surface that expands ventrolaterally to form the posterior roof of the deep sulcus palatino-pterygoideus, thereby separating the spaces of the orbital cavity anteriorly and temporal cavity posteriorly (Fig. 6). The ridge becomes shallower posteroventrally but continues toward the trigeminal foramen, where it levels off just before reaching it. This ridge is strongly developed in OMNH 66106 and more pronounced and extended than that of Arundelemys dardeni (Evers, Rollot & Joyce, 2021), Pleurosternon bullockii (Evers, Rollot & Joyce, 2020), and Uluops uluops (Rollot, Evers & Joyce, 2021a). A similar ridge is present in Gamerabaena sonsalla (Lyson & Joyce, 2010), but it extends far more posteroventrally, with an extended contribution of the quadrate to it. Anteroventrally, the descending process of the parietal contacts the pterygoid and epipterygoid (Fig. 7). On the right side, the suture between the parietal and prootic is easily traceable in the µCT image stack, allowing us to confidently reconstruct that area. There, the parietal sends a process along the posterodorsal margin of the trigeminal foramen (for cranial nerve V) that contacts the pterygoid posteroventrally and prevents the prootic from contributing externally to the margin of the trigeminal foramen (Fig. 7B). Such a process of the parietal is present in many paracryptodires, and either fully (e.g., Uluops uluops: Rollot, Evers & Joyce, 2021a; Rollot, Evers & Joyce, 2021b; Pleurosternon bullockii: Evers, Rollot & Joyce, 2021) or partially (e.g., Glyptops ornatus: Gaffney, 1979) excludes the prootic from contributing to the margin of the trigeminal foramen. The morphology of OMNH 66106 is quite different, in that the parietal process only forms a superficial cover posterior to the trigeminal foramen. If the parietal is digitally disarticulated (Fig. 7C), the trigeminal foramen remains fully surrounded by portions of the prootic and pterygoid. Thus, the prootic and pterygoid form the internal margin of the trigeminal foramen (visible in an internal, medial view onto the inner braincase surface, or when the parietal is disarticulated as in Fig. 7C), but the external margin is formed by the parietal and pterygoid to the exclusion of the prootic (Fig. 7B). On the left side, our reconstructions show that the prootic forms the posterodorsal margin of the trigeminal foramen. This area, however, is poorly preserved in the µCT image stack, which led to segmentation uncertainties on that side of the skull. We, therefore, consider the reconstructions on the right side as correct and dismiss those on the left for further anatomical discussions, and suggest that the prootic is excluded from contributing to the lateral margin of the trigeminal foramen in OMNH 66106. Within the upper temporal fossa, the descending process contacts the supraoccipital dorsally and the prootic posterolaterally.

Figure 6 Three-dimensional renderings of the anterior area of the braincase of OMNH 66106.

(A) Anterolateral view of the right parietal, right prootic, right pterygoid, and supraoccipital. (B) Ventral view of the parietals. Abbreviations: ap-pt, anterior process of the pterygoid; dp-pa, descending process of the parietal; pa, parietal; pro, prootic; pt, pterygoid; r-spp, ridge posteriorly framing the sulcus palatino-pterygoideus; so, supraoccipital; spp, sulcus palatine-pterygoideus; s-so, suture with supraoccipital; vf, vertical flange of the external process of the pterygoid.

Figure 7 Three-dimensional renderings of the right trigeminal foramen of OMNH 66106.

(A) Right lateral view of OMNH 66106 showing the location of the area of interest. (B) Close-up on the right trigeminal foramen area highlighting its external margin. (C) Close-up on the right trigeminal foramen area showing its internal margin. The margins of the trigeminal foramen are highlighted by the dashed circles. Abbreviations: epi, epipterygoid; fnt, foramen nervi trigemini; pa, parietal; pro, prootic; pt, pterygoid.

Postorbital

Both postorbitals are preserved in OMNH 66106 but are damaged (Figs. 3A, 3C and 3D). The dorsal surface along the anterior part of the left postorbital is eroded and partially broken on the right side, and a center piece of the right postorbital is completely missing (Fig. 3A). The anterior part of the postorbital forms a ventral process that is medially expanded to form the posterior wall of the fossa orbitalis along with the medial process of the jugal (Fig. 3E). This ventral process of the postorbital rests on the jugal, and its medial aspect forms the septum orbitotemporale, which separates the orbit from the temporal cavity. The ventral process of the postorbital is thus similar to that of Arundelemys dardeni (Evers, Rollot & Joyce, 2021), Pleurosternon bullockii (Evers, Rollot & Joyce, 2020), and Uluops uluops (Rollot, Evers & Joyce, 2021a). Anteroventrally, the ventral process has a short contact with the maxilla (Figs. 3C and 3D), which prevents the jugal from contributing to the orbit margin, as is also the case in Arundelemys dardeni (Lipka et al., 2006; Evers, Rollot & Joyce, 2021), Arvinachelys goldeni (Lively, 2015), Glyptops ornatus (Gaffney, 1979), and Pleurosternon bullockii (Evers, Rollot & Joyce, 2020), and the baenodds Boremys pulchra (Brinkman & Nicholls, 1991), Plesiobaena antiqua (Brinkman, 2003), Gamerabaena sonsalla (Lyson & Joyce, 2010), and Saxochelys gilberti (Lyson, Sayler & Joyce, 2019). No contact between the ventral process of the postorbital and pterygoid is apparent in OMNH66106, as is it prevented by the medial process of the jugal. Similarly, a contact with the palatine along the foot of the septum orbitotemporale is unclear. In dorsal view, the postorbital contacts the frontal anteromedially, the parietal medially, the squamosal posteriorly, the quadratojugal posteroventrally, and the jugal ventrally (Figs. 3A and 3C–3E). Posterior to its ventral process, the postorbital is developed as a relatively thin, plate-like bone, that slightly narrows towards the posterior end. The contact between the parietal and squamosal prevents the postorbital from contributing to the upper temporal emargination (Fig. 3A), as in Dorsetochelys typocardium (Evans & Kemp, 1976), Helochelydra nopcsai (Joyce et al., 2011), Naomichelys speciosa (Joyce, Sterli & Chapman, 2014), Pleurosternon bullockii (Evans & Kemp, 1975; Evers, Rollot & Joyce, 2020), and Uluops uluops (Rollot, Evers & Joyce, 2021a), and the baenids Baena arenosa, Chisternon undatum, and Neurankylus torrejonensis (Gaffney, 1972a; Lyson et al., 2016).

Jugal

The jugal forms a lateral vertical plate and a medially expanded process (Figs. 3C and 3E). The anterior aspect of this process is exposed within the fossa orbitalis where it contacts the maxilla anteriorly, the palatine laterally, and the ventral process of the postorbital dorsally. Our reconstructions of the jugal and maxilla differ between the right and left sides of the skull in that exposure of the right jugal is greater than that of the left jugal. This difference is ultimately minor and does not affect the contributions of either bone to any cranial structure, but as the right side of OMNH 66106 is better preserved than the left, we consider the reconstructions of the right jugal and maxilla as being the most plausible ones. With the ventral process of the postorbital, this medial process of the jugal contributes to the formation of the mediolaterally expanded posterior wall of the orbit, the septum orbitotemporale (Fig. 3E). Posterior to that wall, the medial process of the jugal is exposed within the temporal cavity and contacts the external process of the pterygoid posteriorly.

The vertical plate of the jugal forms the anterodorsal margin of the moderately deep cheek emargination, which just reaches the level of the lower margin of the orbit (Figs. 3C and 3D). The jugal does not contribute to the labial ridge, as evidenced by a slight posterior extension of the maxilla just beneath the jugal that is visible along the anterior margin of the cheek emargination (Figs. 3C and 3D). This condition is similar to that of most paracryptodires, with the exception of Compsemys victa (Lyson & Joyce, 2011) and Gamerabaena sonsalla (Lyson & Joyce, 2010), in which such a contribution is present. Anteriorly, the contact between the postorbital and maxilla along the posteroventral corner of the orbit prevents the jugal from contribution to the margin of the latter (Figs. 3C and 3D). Below that contact, the jugal contacts the maxilla along an S-shaped suture. The vertical plate of the jugal otherwise tapers posteriorly to contact the postorbital dorsally and the quadratojugal posteriorly (Fig. 3D).

Quadratojugal

The right quadratojugal is preserved in its original position and in contact with the surrounding bones, but its left counterpart is displaced medially within the temporal fossa (Figs. 3C and 3D). The quadratojugal is a thin triradiate element that forms the posterodorsal margin of the cheek emargination and contacts the jugal anteriorly, the postorbital dorsally, the squamosal posterodorsally, and the quadrate posteriorly (Figs. 3B and 3D). The anterior process of the quadratojugal tapers below the posterior part of the vertical plate of the jugal (Fig. 3D). The posterodorsal process of the quadratojugal is well developed and extends dorsally above the cavum tympani between the quadrate and the postorbital (Figs. 3C and 3D), as in most paracryptodires for which this area is known (Evans & Kemp, 1975; Evans & Kemp, 1976; Gaffney, 1979; Gaffney, 1982; Perez-Garcia et al., 2021) with the exception of some palatobaenines (Brinkman, 2003; Lyson & Joyce, 2009a; Lyson & Joyce, 2009b) and Compsemys victa (Lyson & Joyce, 2011). The posteroventral process of the quadratojugal extends along the anteroventral margin of the quadrate but does not reach the mandibular condyle (Figs. 3C and 3D). The posteriormost aspect of the quadratojugal ends very close to the cavum tympani but the anterior margin of the latter is entirely delimited by the quadrate (Figs. 3C and 3D).

Squamosal

Both squamosals are nearly completely preserved and offer insights into a skull region that often is damaged in fossils. The squamosal forms the posterodorsolateral aspect of the skull, the posterolateral margin of the upper temporal emargination, the posterodorsal margin of the cavum tympani, and most of the deep antrum postoticum (Figs. 3A, 3C, 3D and 3F). Along the skull roof, the squamosal contacts the posterodorsal process of the quadratojugal anterolaterally, the postorbital anteriorly, and the parietal anteromedially (Figs. 3A, 3C and 3D). Posteriorly, the squamosal forms a cap-like structure that contains the deep and broad antrum postoticum internally. The posterior tip of this cap forms relatively short processes that do not extend posteriorly beyond the level of the paroccipital process, and extend only little beyond the crista supraoccipitalis (Figs. 3A and 3F). Medially, the squamosal forms a sharp margin that finishes the upper temporal emargination posteriorly. This margin is medially expanded over the temporal fossa, forming a deep recess in the lateral wall of the fossa between the squamosal posteriorly, and squamosal and quadrate more anteriorly (Fig. 3F). The posterior surface of the squamosal is dominated by two structures. Medioventrally, there is a low but robust ridge leading to the posterior end of the dorsoventrally flattened paroccipital process of the opisthotic. Posteroventrolaterally, the ridge defines a concavity for the musculus depressor mandibulae (Gaffney, 1982), which is jointly formed by the squamosal and a posterodorsal process of the quadrate above the level of the incisura columella auris. Within the temporal fossa, the squamosal contacts the quadrate anteriorly and anteromedially and the paroccipital process of the opisthotic posteromedially (Figs. 3A and 3F).

Premaxilla

Neither premaxilla is preserved in OMNH 66106.

Maxilla

Both maxillae are damaged in OMNH 66016. The right maxilla lacks its anteriormost aspect along the apertura narium externa and the area of contact with the premaxilla (Figs. 3A, 3B, 3D, and 3E). The left maxilla is almost completely missing and only its posteriormost aspect along the posteroventral margin of the orbit and a small piece of bone that contacts the dorsal plate of the prefrontal medially are preserved (Figs. 3A–3C and 3E). The preserved portions nevertheless allow assessing most contacts and the contributions of the maxillae to cranial structures. The maxilla forms the lateral margins of the apertura narium externa, the anterolateral wall of the fossa nasalis, the anterior and ventral margins of the orbit, and the anteroventral margin of the cheek emargination (Figs. 3A–3E). The ascending process of the maxilla extends dorsally along the lateral aspects of the prefrontal and is slightly exposed on the skull roof (Figs. 3A and 3C–3E). The ascending process contacts the nasal dorsomedially on the right side, but such a contact is prevented on the left by an unusual thin, anteriorly directed process of the prefrontal that reaches the margin of the apertura narium externa (see Nasal above; Figs. 3A, 3C and 3E). Within the orbital fossa, the maxilla forms the lateral margin of the foramen orbito-nasale and contacts the palatine medially posterior to the foramen orbito-nasale, and the jugal posteriorly (Fig. 3A). The external suture seam with the jugal is V-shaped on the right side but almost straight on the left (see Jugal above for a short discussion about this difference). In lateral view, the maxilla contacts the postorbital posterodorsally, which prevents the jugal from contributing to the orbit, and the jugal posteriorly along an S-shaped suture (Figs. 3C and 3D). The triturating surface of the right maxilla is relatively narrow, and slightly expands posteriorly (Fig. 8A), reminiscent of that of Arundelemys dardeni (Lipka et al., 2006; Evers, Rollot & Joyce, 2021), Pleurosternon bullockii (Evans & Kemp, 1975; Evers, Rollot & Joyce, 2020), and Uluops uluops (Rollot, Evers & Joyce, 2021a), but differing from the notably broader triturating surfaces found in Eubaena cephalica (Rollot, Lyson & Joyce, 2018), Goleremys mckennai (Hutchison, 2004), Palatobaena spp. (Archibald & Hutchison, 1979; Lyson & Joyce, 2009a), Saxochelys gilberti (Lyson, Sayler & Joyce, 2019), and Stygiochelys estesi (Gaffney, 1972a). The triturating surface of OMNH 66016 is transversely slightly concave and is laterally bordered by a high labial ridge (Figs. 8B and 8C). The labial ridge is preserved along the posterior part of the right maxilla. The labial ridge of OMNH 66106 was likely curved, but we are not able to determine the exact extent of it as about half of the maxilla is missing anteroventrally. The foramen supramaxillare is visible on the left side and located at about mid-length of the preserved portion of the left maxilla, just lateral to the suture with the palatine. As both maxillae are damaged, only small and fragmentary portions of the canalis alveolaris superior can be identified within the latter.

Figure 8 Three-dimensional renderings of the right maxilla and palatine of OMNH 66106.

(A) Ventral view of the right maxilla and palatine. (B) Anterior view of the right maxilla. (C) Posterior view of the right maxilla. Dashed lines highlight the limits of the triturating surface of the maxilla in (A) and the extent of the labial margin in (B) and (C). Abbreviations: ap-mx, ascending process of the maxilla; fpp, foramen palatinum posterius; lr, labial ridge; mx, maxilla; pal, palatine; s-ju, suture with the jugal; ts, triturating surface.

Vomer

The vomer is not preserved in OMNH 66016.

Palatine

Both palatines are preserved in OMNH 66106 but are slightly displaced from their original position (Fig. 3B). In addition, the left palatine is damaged, missing its medial- and anteriormost portions (Fig. 3B). The palatine forms the posterior margin of the foramen orbito-nasale, the posteromedial margin of the fossa orbitalis, and forms the entire foramen palatinum posterius (Fig. 8A), as in Uluops uluops (Rollot, Evers & Joyce, 2021a) and the baenids Boremys pulchra (Brinkman & Nicholls, 1991), Cedrobaena putorius (Lyson & Joyce, 2009b), Chisternon undatum (Gaffney, 1982), Eubaena cephalica (Rollot, Lyson & Joyce, 2018), Hayemys latifrons (Gaffney, 1982), and Stygiochelys estesi (Gaffney, 1982). The palatine contacts the maxilla laterally, the jugal posterolaterally, and the pterygoid posteriorly (Figs. 3A, 3B and Fig. 8A). As the vomer is missing in OMNH 66106, the determination of the presence or absence of a contact between the latter and the palatine is not directly possible. However, a few clues show that such a contact was present in this specimen. In particular, the medial portion of the right palatine is nearly complete and its medial surface seems to correspond to an articular surface (Figs. 3B and 8A). Although the medialmost portion of the left palatine is incomplete, if it were, it would not reach the medial articular surface of its counterpart and a relatively large space would remain between the two palatines (Fig. 3B). Furthermore, no paracryptodire is known to have a medial contact between the palatines as the vomer always posteriorly contacts the pterygoids and fully separates the palatines from contacting one another. We are therefore confident in speculating that the palatine medially contacted the vomer for most of its length and that the palatines did not contact one another along the midline. A contact with the prefrontal was likely present, but we are not able to observe it as some shearing has displaced the bones.

Pterygoid

The pterygoids are preserved and mostly complete in OMNH 66106, only lacking the anteriormost part of the anterior process (Fig. 3B). The pterygoid bears an anteriorly directed process that would have contacted the posterior portion of the vomer anteriorly (see Palatine above for further explanation about inferred contacts for the palatine and vomer) and, along with its counterpart that it contacts medially, prevents the palatines from contacting each other along the midline (Fig. 3B). A similar anterior process of the pterygoid is found in Arundelemys dardeni (Evers, Rollot & Joyce, 2021), Dorsetochelys typocardium (Evans & Kemp, 1976), and Uluops uluops (Rollot, Evers & Joyce, 2021a), and appears to differ from other known paracryptodires (Gaffney, 1972a; Gaffney, 1979; Evers, Rollot & Joyce, 2020; Lyson & Joyce, 2011). Anterolaterally, the pterygoid forms a well-developed processus pterygoideus externus with a vertical flange (Figs. 3C, 3D and 6A), similar to that of Arundelemys dardeni (Evers, Rollot & Joyce, 2021), Arvinachelys goldeni (Lively, 2015), Glyptops ornatus (Gaffney, 1979), Plesiobaena antiqua (Brinkman, 2003), Pleurosternon bullockii (Evers, Rollot & Joyce, 2020), Pleurosternon moncayensis (Perez-Garcia et al., 2021), Saxochelys gilberti (Lyson, Sayler & Joyce, 2019), Uluops uluops (Rollot, Evers & Joyce, 2021a), and also to the processus trochlearis pterygoidei of pleurodires, but which differs from the reduced processus pterygoideus externus of most palatobaenines (Archibald & Hutchison, 1979; Hutchison, 2004; Lyson & Joyce, 2009a; Lyson & Joyce, 2010; Lyson et al., 2021). The processus externus pterygoideus anteriorly contacts the medial process of the jugal along an interdigitated joint, making the process well integrated with the septum orbitotemporale, again as in many pleurodires.

Along its dorsal surface, the pterygoid contacts the descending process of the parietal anteriorly, the epipterygoid laterally, and the prootic and opisthotic posteriorly along butt or slightly interdigitated joints (Fig. 7). The contacts with the epipterygoid and descending process of the parietal in turtles are usually developed along a dorsally raised crest of the pterygoid, the crista pterygoidei. This crista is notably low in paracryptodires generally, a feature that has not yet been explicitly mentioned before, but which can be appreciated particularly well in Arundelemys dardeni (Evers, Rollot & Joyce, 2021) and Uluops uluops (Rollot, Evers & Joyce, 2021a) due to their well-preserved pterygoids. In OMNH 66106, the crista pterygoidei appears low even by comparison with these of other paracryptodires and is only developed clearly over a short distance just anterior to the trigeminal foramen. In this region, the pterygoid forms the posteroventral margin of the foramen nervi trigemini (Fig. 7). Anterior to the foramen, the lateral surface of the low crista pterygoidei serves as the contact surface for the epipterygoid, just as in Arundelemys dardeni (Evers, Rollot & Joyce, 2021) and Uluops uluops (Rollot, Evers & Joyce, 2021a). Unlike in Arundelemys dardeni, but as in Uluops uluops, the epipterygoid remains excluded from the trigeminal foramen (Fig. 7). The crista pterygoidei is laterally inclined in OMNH 66106, which is not observed in other paracryptodires. It is unclear if this feature is caused by slight dorsoventral compression of the fossil, but the well-articulated trigeminal region on the left side points to the possibility that this deflection is original. As a consequence, the sulcus cavernosus medial to the crista pterygoidei is broader than in Arundelemys dardeni and particularly Uluops uluops. In addition, the ‘sulcus’ is not really excavated to a channel in OMHN 66106, but is notably flat. This appearance is accentuated by the low trabeculae of the rostrum basisphenoidale of the parabasisphenoid, which usually create more topology between right and left sulcus cavernosus and the centrally placed anterior end of the sella turcica (see Uluops uluops for this more ‘conventional’ morphology; Rollot, Evers & Joyce, 2021a).

The parabasisphenoid is exposed on the ventral surface of the skull for most of its length and the pterygoid has an elongate contact with the latter medially (Fig. 3B). The contact between the pterygoid and its counterpart is thus limited to the anteriormost portion of that bone, mainly along the anterior processes. Posteriorly, the pterygoid bears a well-developed posterior process that laterally contacts the quadrate and has a notably extensive contact medially with the basioccipital (Fig. 3B). Such an extensive contact with the basioccipital is similar to the condition found in Arundelemys dardeni (Lipka et al., 2006), Neurankylus eximius (Brinkman & Nicholls, 1993), and baenodds (Gaffney, 1982; Hutchison, 2004; Joyce & Lyson, 2015; Lyson, Sayler & Joyce, 2019; Lyson et al., 2021), but contrasts with that of Dorsetochelys typocardium (Evans & Kemp, 1976), Glyptops ornatus (Gaffney, 1979), Pleurosternon bullockii (Evans & Kemp, 1975), Pleurosternon moncayensis (Perez-Garcia et al., 2021), and Uluops uluops (Rollot, Evers & Joyce, 2021a), in which the posterior process of the pterygoid does not extend beyond the posterior limit of the parabasisphenoid. It is not clear if the pterygoid also contacted the exoccipital in this region because the sutures between the basioccipital and exoccipitals are unclear. The posterior margin of the pterygoid is strongly concavely notched between its contacts with the basioccipital and quadrate. The pterygoid fossa on the ventral surface of the posterior process is extremely shallow, which constitutes a noteworthy difference to pleurosternids but also other early branching baenids such as Arundelemys dardeni (Evers, Rollot & Joyce, 2021) and Trinitichelys hiatti, which have relatively deep pterygoid fossae.

Figure 9 Carotid circulation and vidian canal system of OMNH 66106.

Three-dimensional renderings of the parabasisphenoid, pterygoids, and canals of the circulatory system in (A) dorsal and (B) ventral view. (C) Ventral view of the segmented skull showing the location the area of interest highlighted in (D) and (E). (D) Close-up on the basicranium highlighting the arteries and nerves of the circulation system. (E) Close-up on the basicranium highlighting the foramina for the circulation system. Abbreviations: ac, arteria carotis cerebralis; ap, arteria palatina; ccb, canalis caroticus basisphenoidalis; ccl, canalis caroticus lateralis; ccv, canalis cavernosus; cnf, canalis nervus facialis; cnv, canalis nervus vidianus; faccb, foramen anterius canalis carotici basisphenoidalis; fpccb, foramen posterius canalis carotici basisphenoidalis; fpccl, foramen posterius canalis carotici lateralis; fpcnv, foramen posterius canalis nervi vidiani; pbs, parabasisphenoid; pt, pterygoid; vn, vidian nerve.

The pterygoid is also involved in the formation of various structures related to the blood circulation and innervation systems (Fig. 9). Along its dorsal surface and lateral to the rostrum basisphenoidale, the pterygoid forms the sulcus cavernosus, which served as the passage for the lateral head vein posteriorly through the braincase (Gaffney, 1972b). The pterygoid forms most of the foramen cavernosum, which is dorsally bordered by the prootic, and the ventral margin of the canalis cavernosus (Fig. 9A). In the area at about mid-length along the parabasisphenoid-pterygoid suture, we identify three foramina that are formed by either the parabasisphenoid, the pterygoid, or both bones (Figs. 9B and 9E). These foramina seem to be located within a shallow depression (Fig. 9E), but a true carotid pit or a carotid groove, as that found in Arundelemys dardeni (Evers, Rollot & Joyce, 2021) and Uluops uluops (Rollot, Evers & Joyce, 2021a), are not visible in OMNH 66106. We identify these foramina as structures related to the circulatory system, as recently described (see Rollot, Evers & Joyce, 2021a; Rollot, Evers & Joyce, 2021b), and we provide our rationale about their identification below. The anteriormost of the three foramina is bordered by the parabasisphenoid medially and the pterygoid laterally, and leads into a canal that extends anteriorly along the parabasisphenoid-pterygoid suture (Figs. 9A, 9B, 9D and 9E). Although the anterior part of this canal and its anterior exit foramen are not clearly visible in the µCT scans, the location of the posterior portion of this canal along the parabasisphenoid-pterygoid suture is consistent with the expectations for a palatine artery canal, thus allowing us to identify it as the canalis caroticus lateralis (Rollot, Evers & Joyce, 2021b). The first foramen thus corresponds to the foramen posterius canalis carotici lateralis. The second foramen of interest is located slightly posterolateral to the foramen posterius canalis carotici lateralis and is entirely formed by the pterygoid (Figs. 9B, 9D and 9E). The foramen leads into a canal that is directed anteriorly and crosses the pterygoid for all its length (Figs. 9A and 9B). An anterior exit foramen is not clearly apparent but is very likely located on the dorsal surface of the pterygoid, close to the ventral limit of the descending process of the parietal. The canal that crosses the pterygoid is fully separated from the canalis caroticus lateralis, and its location within the pterygoid, along with that of its posterior foramen, is typical of the canal for the vidian nerve (Rollot, Evers & Joyce, 2021a; Rollot, Evers & Joyce, 2021b). We therefore respectively identify the foramen and canal as the foramen posterius canalis nervi vidiani and canalis nervus vidianus. The third foramen of interest is located posteromedial to the foramen posterius canalis nervi vidiani (Figs. 9B, 9D and 9E). The foramen is largely formed by the parabasisphenoid but laterally bordered by the pterygoid, and the associated canal extends anteromedially through the parabasisphenoid to join the sella turcica (Fig. 9A). This arrangement is that of a canal for the cerebral artery (Rollot, Evers & Joyce, 2021a; Rollot, Evers & Joyce, 2021b), and the foramen and canal are thus confidently identified as the foramen posterius canalis carotici basisphenoidalis and canalis caroticus basisphenoidalis, respectively. The arrangement of the aforementioned foramina along the ventral surface of the skull, as well as the presence of a canalis caroticus lateralis, is strikingly similar to the circulatory system described for Uluops uluops (Rollot, Evers & Joyce, 2021a), although the surface topology of the pterygoid surrounding these foramina varies between both taxa.

Epipterygoid

Both epipterygoids are preserved in OMHN 66106. The epipterygoid is small (Fig. 7), much smaller than that of Arundelemys dardeni (Evers, Rollot & Joyce, 2021) and Uluops uluops (Rollot, Evers & Joyce, 2021a). The epipterygoid of OMNH 66106 is a small, triangular element located just anteroventral to the foramen nervi trigemini, but it does not truly contribute to the ventral margin of the latter (Fig. 7). The epipterygoid contacts the descending process of the parietal anterodorsally and anteriorly, and the pterygoid ventrally and posteriorly (Fig. 7). An ossified epipterygoid is generally present in non-baenodd paracryptodires (Evans & Kemp, 1975; Evans & Kemp, 1976; Evers, Rollot & Joyce, 2020; Evers, Rollot & Joyce, 2021; Gaffney, 1979; Lipka et al., 2006; Perez-Garcia et al., 2021; Rollot, Evers & Joyce, 2021a). Due to the small size of the element in OMHN 66106 and its currently hypothesized phylogenetic position (see Joyce, Rollot & Cifelli, 2020), we propose that the ‘absence’ of the epipterygoid in baenodds may be a full reduction of the element, rather than a fusion with the surrounding bones of the braincase, particularly the pterygoid. This is possibly contradicted by data presented by Brinkman (2003), who found an epipterygoid to be present in small individuals of Plesiobaena antiqua. However, the reported epipterygoid is extremely small and does not show the typical epipterygoid shape, as it is retracted from the trigeminal foramen and also limited to a small area along the sutural contact of quadrate, prootic and pterygoid. This observation may point to the possibility that the epipterygoid in baenodds is indeed evolutionarily reduced, and that small remnants of the bone are best identified in juveniles.

Quadrate

The quadrates are almost completely preserved, with some damage that affects the left mandibular condyle (Fig. 3B). The quadrate forms the cavum tympani, the ventral part of the antrum postoticum, the lateral portion of the cavum acustico-jugulare, and the incisura columella auris (Figs. 3C, 3D and 3F). In lateral view, the quadrate contacts the quadratojugal anteriorly and anterodorsally along a convex suture and the squamosal posterodorsally (Figs. 3C and 3D). The cavum tympani is deep but slightly smaller than the orbit, and is separated from the large antrum postoticum by a shallow ridge. The antrum postoticum is extensive, and, unusually for turtles, also excavates anteriorly into the quadrate. This excavation creates a clear separation of two cavities within the cavum tympani, in the form of a bowl-like lateral part, and the deep cavern formed by the antrum postoticum. This anterior antrum expansion is not visible in Arundelemys dardeni (Lipka et al., 2006; Evers, Rollot & Joyce, 2021), but the region is very similar in Trinitichelys hiatti. Uluops uluops and Pleurosternon bullockii again show a similar morphology (Rollot, Evers & Joyce, 2021a; Evers, Rollot & Joyce, 2020), although the similarity is less conspicuous than with Trinitichelys hiatti. The incisura columella auris is posteriorly open (Figs. 3C and 3D). Posterior to the incisura, the quadrate has a posterior process that buttresses the squamosal and forms a posterolaterally and slightly ventrally exposed fossa with it (see Squamosal).

Within the lower temporal fossa, the quadrate contacts the pterygoid anteromedially and bears a short epipterygoid process, which is better preserved on the right side. Within the upper temporal fossa, the quadrate contacts the prootic anteromedially, the opisthotic posteromedially, and the squamosal posterolaterally (Fig. 3A). A contact with the supraoccipital is prevented by a short contact between the prootic and opisthotic just posterior to the foramen stapedio-temporale (Fig. 3A), as is also the case in Arundelemys dardeni (Lipka et al., 2006; Evers, Rollot & Joyce, 2021) and Pleurosternon moncayensis (Perez-Garcia et al., 2021). The quadrate forms the lateral margin of the foramen stapedio-temporale and canalis stapedio-temporalis, and the lateral half of the processus trochlearis oticum, which is mostly developed in the form of an osseous ridge that overhangs the lower temporal fossa and appears to be slightly more pronounced than that of Arundelemys dardeni (Evers, Rollot & Joyce, 2021), Pleurosternon bullockii (Evers, Rollot & Joyce, 2020), and Uluops uluops (Rollot, Evers & Joyce, 2021a; Rollot, Evers & Joyce, 2021b). Within the cavum acustico-jugulare, the quadrate contacts the paroccipital process of the opisthotic dorsomedially, the prootic anteromedially, and the posterior process of the pterygoid ventromedially (Fig. 3F), and forms the dorsolateral and lateral margins of the aditus canalis stapedio-temporalis. The right mandibular condyle is small and low, only lacking its ventral surface that is eroded off, while the left mandibular condyle is completely missing (Fig. 3B).

Prootic

Both prootics are preserved in OMNH 66106. Within the upper temporal fossa, the prootic contacts the descending process of the parietal anteromedially, the supraoccipital posteromedially, the opisthotic posteriorly just posterior to the foramen stapedio-temporale, and the quadrate posterolaterally (Fig. 3A). The prootic forms the medial half of the processus trochlearis oticum and the medial margin of the foramen stapedio-temporale (Fig. 3A) and canalis stapedio-temporalis. The prootic is prevented from contributing to the external margin of the foramen nervi trigemini on the right side by a posteroventral process of the parietal that contacts the pterygoid along the posterior margin of the aforementioned foramen (Fig. 7B). On the left side, we are unable to distinguish the prootic, opisthotic, and quadrate from one another and thus segmented the three bones as a single element. A portion of the segmented block appears to form the posterodorsal margin of the foramen nervi trigemini but the sutures are not fully clear in that area, and the portion of bone that is exposed along the margin of the aforementioned foramen might even possibly belong to the parietal instead. Thus, we think that the morphology on the right side is more accurate. Although the prootic is excluded from the external opening of the trigeminal foramen, this arrangement is different on the internal side: here, the prootic forms a slightly hooked process that extends anterior to the level of the foramen cavernosum. The process reaches and contacts the crista pterygoidei in the anterior margin of the trigeminal foramen, so that the foramen is fully enclosed by the prootic and pterygoid, lacking participation of the parietal (or epipterygoid; Fig. 7C). Only when the parietal model is switched on in the digital models, it overlays the lateral surface of the prootic completely, thus excluding the prootic from the foramen externally (Fig. 7B). The ventral surface of the hooked process is concavely excavated, so that the prootic forms a well-developed (but mediolaterally narrow) cavum epiptericum. Below the hooked process, the prootic contributes to the formation of the foramen cavernosum dorsally and the canalis cavernosus medially and dorsally. The prootic also forms the anterior margin of the hiatus acusticus, the anterior part of the cavum labyrinthicum, the anterior half of the canalis semicircularis anterior and horizontalis, and the anterior margin of the large fenestra ovalis, which is ventrally closed by a prootic-opisthotic contact and is thus fully circumscribed by these two bones, as is also the case in Arundelemys dardeni (Evers, Rollot & Joyce, 2021) and Pleurosternon moncayensis (Perez-Garcia et al., 2021). The posterior surface of the prootic just lateral to the fenestra ovalis is excavated by a small pericapsular recess fossa, as in Arundelemys dardeni (Evers, Rollot & Joyce, 2021) but also pleurosternids (Evers, Rollot & Joyce, 2020). The canals for the undivided facial (VII) and acoustic nerves (VIII) are fully enclosed in the prootic. The canalis nervus facialis is partially preserved on both sides and extends laterally from the fossa acustico-facialis to approach the canalis cavernosus closely (Fig. 9A). The poor preservation of that area, however, does not allow us to observe a contact between the canalis nervus facialis and canalis cavernosus. As the split of the facial nerve into the hyomandibular and vidian nerves is not apparent along the visible portion of the facial nerve canal, it is likely that this split occurred slightly more laterally, at the point of contact between the canalis nervus facialis and canalis cavernosus, which is not preserved in OMNH 66106. From there, the vidian nerve likely extended ventrally by means of the canalis pro ramo nervi vidiani to join the ventral surface of the skull, and then anteriorly re-entered the skull through the foramen posterius canalis nervi vidiani into the canalis nervus vidianus, two osteological correlates we have identified with confidence (see Pterygoid; Figs. 9B, 9D and 9E). The pattern of facial and vidian innervation in OMNH 66106 was thus probably identical to that inferred for Arundelemys dardeni (Evers, Rollot & Joyce, 2021), Pleurosternon bullockii (Evers, Rollot & Joyce, 2020), Pleurosternon moncayensis (Perez-Garcia et al., 2021), and Uluops uluops (Rollot, Evers & Joyce, 2021a). Within the cavum acustico-jugulare, the prootic contacts the opisthotic posterodorsally, the quadrate posterolaterally, and the pterygoid posteroventrally, and forms the dorsomedial margin of the aditus canalis stapedio-temporalis.

Opisthotic

Both opisthotics are well preserved, with the exception of the processus interfenestralis, which is damaged on both sides, albeit better preserved on the left. As sutures with the prootic and quadrate were not clear, these bones were segmented as a single model for the left side (Fig. 3A). Within the upper temporal fossa, the anterior part of the opisthotic contacts the supraoccipital anteromedially, the prootic anteriorly, and the quadrate anterolaterally, as documented by well traceable sutures on the right side (Fig. 3A). The posterior portion of the opisthotic forms the paroccipital process, which contacts the basioccipital-exoccipitals complex medially and the squamosal laterally, and borders the foramen jugulare anterius anteriorly (Figs. 3A and 3F). The posterior margin of the paroccipital process is developed as a dorsoventrally flat ridge. The paroccipital process also roofs the cavum acustico-jugulare and forms the dorsolateral margin of the fenestra postotica. A well-defined ridge mediolaterally crosses the dorsal surface of the posterior part of the paroccipital process, as is the case in Uluops uluops (Rollot, Evers & Joyce, 2021a) but not in Arundelemys dardeni (Evers, Rollot & Joyce, 2021). The opisthotic does not contribute to the foramen stapedio-temporale or canalis stapedio-temporalis (Fig. 3A). The opisthotic forms the posterior margin of the hiatus acusticus, the posterior part of the cavum labyrinthicum, and the posterior half of the canalis semicircularis horizontalis and posterior. The left processus interfenestralis is almost completely preserved, separates the cavum labyrinthicum from the recessus scalae tympani, and forms the lateral margin of the fenestra perilymphatica, the anterolateral margin of the recessus scalae tympani, and the posterior margin of the fenestra ovalis. The ventral part of the processus interfenestralis is developed into a relatively large footplate that contacts the prootic anteriorly, the basioccipital-exoccipitals complex ventromedially, and the pterygoid ventrally and ventrolaterally. The foramen externum nervi glossopharyngei and foramen internum nervi glossopharyngei (IX) are not preserved and a contact with the parabasisphenoid was likely absent.

Supraoccipital

The supraoccipital is almost completely preserved and forms the dorsal margin of the cavum cranii and foramen magnum, the medial margin of the upper temporal fossa, and the posteromedial margin of the upper temporal emargination (Figs. 3A and 3F). Within the upper temporal fossa, the supraoccipital contacts the descending process of the parietal anteriorly, the prootic anterolaterally, the opisthotic laterally, and the basioccipital-exoccipitals complex posterolaterally (Figs. 3A and 3F). A lateral contact with the quadrate is prevented by the contact between the prootic and opisthotic (see Quadrate above; Fig. 3A). The crista supraoccipitalis is mediolaterally thin, dorsoventrally relatively short, and likely did not extend posteriorly beyond the level of the occipital condyle (Figs. 3A and 3F). The posteromedial part of the parietals cover about the anterior half of the crista supraoccipitalis. Only at their posterior ends, the thinning posterior parietal processes diverge laterally for a small, slightly broadened shelf of the crista supraoccipitalis to form part of the skull roof (Fig. 3A). The missing portion of the crista is indeed small, and the preserved part that is exposed on the skull roof, along with the posteriormost portion of the parietals, clearly show that the skull roof narrows posteriorly to form a pointed tip (Fig. 3A). The crista supraoccipitalis is thus inferred to have been only slightly exposed along the skull roof, as is the case in Palatobaena bairdi (Archibald & Hutchison, 1979) and Plesiobaena antiqua (Brinkman, 2003), which differs from the enlarged exposure of the crista as found in Dorsetochelys typocardium (Evans & Kemp, 1976), Eubaena cephalica (Rollot, Lyson & Joyce, 2018), Cedrobaena brinkman (Lyson & Joyce, 2009b), and Uluops uluops (Rollot, Evers & Joyce, 2021a). The supraoccipital roofs the cavum labyrinthicum and forms about the posterior half of the canalis semicircularis anterior and anterior half of the canalis semicircularis posterior. Anterior to the hiatus acusticus and just dorsal to the supraoccipital-prootic contact, the supraoccipital forms the foramen aquaducti vestibuli.

Basioccipital-exoccipital

The basioccipital and exoccipitals are completely preserved in OMNH 66106. As we are unable to identify the sutures between the exoccipitals and the basioccipital, we segmented the three bones as a single bony complex, which is described as such herein (Figs. 3A, 3B and 3F). The basioccipital-exoccipital-complex forms the posterolateral and posteroventral parts of the cavum cranii, the lateral and ventral margins of the cavum cranii, the lateral margin of the fenestra postotica, the posterior margin of the foramen jugulare anterius, and most of the margin of the recessus scalae tympani (Fig. 3F). Within the cavum cranii, the basioccipital-exoccipital-complex forms a low crista dorsalis basioccipitalis along its dorsal surface. The basioccipital-exoccipital-complex is slightly exposed within the upper temporal fossa, where it contacts the supraoccipital medially and the opisthotic laterally (Fig. 3A). In posterior view, the basioccipital-exoccipital-complex contacts the supraoccipital dorsomedially and the paroccipital process of the opisthotic dorsolaterally (Fig. 3F). Ventrolaterally, the basioccipital-exoccipital-complex forms short and thin tubercula basioccipitale, to which the pterygoid slightly contributes (Figs. 3B and 3F). As the sutures between the exoccipitals and basioccipital are not apparent, we are not able to assess the contribution of either bone to the articular surface of the occipital condyle. Along the roof of the cavum acustico-jugulare, the basioccipital-exoccipital-complex contacts the opisthotic laterally. Along the floor of the cavum acustico-jugulare, the basioccipital-exoccipital-complex contacts the processus interfenestralis of the opisthotic dorsolaterally and the pterygoid laterally. Two foramina nervi hypoglossi are present, as in Dorsetochelys typocardium (Evans & Kemp, 1976), Glyptops ornatus (Gaffney, 1979), and Pleurosternon moncayensis (Perez-Garcia et al., 2021), with the posterior foramen being much larger than the anterior one. These foramina are not exposed on the occipital surface of the bone complex, but the canals are laterally directed, almost into the cavum acustico-jugulare. In ventral view, the basioccipital-exoccipital-complex contacts the parabasisphenoid anteriorly and the posterior process of the pterygoid laterally (Fig. 3B). The ventral surface of the basioccipital-exoccipital-complex is flat (Fig. 3B).

Parabasisphenoid

The parabasisphenoid is preserved in OMNH 66106 (Fig. 3B). The anterior part of the parabasisphenoid forms the rostrum basisphenoidale, which is developed as a thin sheet of bone that is slightly shorter than the posterior portion of the parabasisphenoid. The rostrum basisphenoidale contacts the pterygoid anteriorly and ventrolaterally (Fig. 3B), and forms the medial margin of the inconspicuous sulcus cavernosus with low trabeculae. The rostrum basisphenoidale is exposed along the ventral surface of the skull, giving the parabasisphenoid an anteroposteriorly elongate and mediolaterally narrow appearance in ventral view (Fig. 3B). Such an exposure of the parabasisphenoid ventrally for nearly all its length is similar to the condition found in Glyptops ornatus (Gaffney, 1979), Pleurosternon bullockii (Evers, Rollot & Joyce, 2020), and Pleurosternon moncayensis (Perez-Garcia et al., 2021). In those three taxa, it remains unclear if the parabasisphenoid fully separates the pterygoids, or if a medial contact between the latter is present anteriorly. In OMNH 66106, the anterior end of the rostrum basisphenoidale appears to be complete and is ventrally covered by the base of the anterior process of the pterygoids. The dorsal surface of the rostrum basisphenoidale is posteriorly excavated by the sella turcica, which forms a moderately deep depression in which the widely spaced foramina anterius canalis carotici basisphenoidalis are located posterolaterally (Fig. 9A). As in other paracryptodires, the clinoid processes are inconspicuous, low and short anterior bumps to either side of the dorsum sellae, which forms the anterior margin of the cup-like fossa in the centrum of the dorsal parabasisphenoid surface. Although it is possible that some parts of the processes are broken, comparison among paracryptodires suggests the processes were simply short in all taxa of this group.

The posterior portion of the parabasisphenoid is dorsoventrally thick and forms a deep depression along its dorsal surface. The foramina posterius canalis nervi abducentis are located on that dorsal surface of the parabasisphenoid. On the left side, we are able to identify the canalis nervus abducentis that extends anteriorly within the parabasisphenoid. The anterior portion of the canalis nervus abducentis is not visible in the µCT image stack, and although it likely joins the area posteroventral to the base of the clinoid process, its exact path is unclear. The bony contributions to the foramen anterius canalis nervi abducentis are thus unknown. The posterior portion of the parabasisphenoid contacts the pterygoid ventrolaterally for all its length, the prootic dorsolaterally, and the basioccipital-exoccipital-complex posteriorly. The posterodorsolateral edge of the parabasisphenoid likely formed the ventral margin of the hiatus acusticus. A contact with the opisthotic is not apparent. The dorsal surface of the posterior portion of the parabasisphenoid bears a low ridge posteromedially, which indicates the presence of a crista basis tuberculi basalis, as in Arundelemys dardeni (Evers, Rollot & Joyce, 2021). In ventral view, the parabasisphenoid contacts the pterygoid laterally and the basioccipital-exoccipital-complex posteriorly (Fig. 3A). At about mid-length along the parabasisphenoid-pterygoid suture, the parabasisphenoid forms the medial margin of the foramen posterius canalis carotici lateralis and, posterior to the latter, most of the margin of the foramen posterius canalis carotici basisphenoidalis (Figs. 9B and 9E). A carotid pit and a carotid groove, as found in Arundelemys dardeni (Evers, Rollot & Joyce, 2021) and Uluops uluops (Rollot, Evers & Joyce, 2021a), are absent. We are unable to find any evidence of a basipterygoid process in OMNH 66106, which is thus likely absent, as is also the case in Arundelemys dardeni (Evers, Rollot & Joyce, 2021) and advanced baenids (Gaffney, 1982), but differs from pleurosternids (Evers, Rollot & Joyce, 2020), including Uluops uluops (Rollot, Evers & Joyce, 2021a). The posterolateral surface of the parabasisphenoid is smooth and anterior tubercula basioccipitale are absent (Fig. 3A).

Discussion

Specimen OMNH 66106, the only known skull of Lakotemys australodakotensis, was initially described by Joyce, Rollot & Cifelli (2020), but crushing of the skull and presence of metal oxides hindered a detailed description. Observations originally made only pertain to the general size and shape of the skull, orbits, and temporal emarginations, and the identification of portions of the frontal, postorbital, parietal, and squamosal bones. The reexamination of OMNH 66106 as part of a project that aims to investigate paracryptodiran relationships convinced us that, despite the difficulty of the task, a bone-by-bone segmentation and reconstruction of the skull was possible. The results obtained from the segmentation of OMNH 66106 allowed us to thoroughly describe every bone that is preserved in this specimen, and, therefore, to provide the first detailed insights into the cranial anatomy of Lakotemys australodakotensis. The present description also adds to recent studies that provided extensive µCT-guided redescriptions of paracryptodiran skulls such as for Arundelemys dardeni (Evers, Rollot & Joyce, 2021), Eubaena cephalica (Rollot, Lyson & Joyce, 2018), Pleurosternon bullockii (Evers, Rollot & Joyce, 2020), Pleurosternon moncayensis (Perez-Garcia et al., 2021), and Uluops uluops (Rollot, Evers & Joyce, 2021a). Although the main goal of the present study is to provide the anatomical basis for future phylogenetic investigations of paracryptodiran relationships, we take the opportunity to highlight similarities and differences of Lakotemys australodakotensis with other paracryptodires (see Table 3 for a summary).

Table 3 Comparative table summarizing the combination of features shared by Arundelemys dardeni and Lakotemys australodakotensis, typical of either Pleurosternidae or Baenidae.

Character	Pleurosternon bullockii	Uluops uluops	Glyptops ornatus	Arundelemys dardeni	Lakotemys australodakotensis	Hayemys latifrons	Eubaena cephalica	Cedrobaena putorius	
Anterior process of the pterygoid	Present	Present	–	Present	Present	Absent	Absent	Absent	
Processus pterygoideus externus	Well-developed	Well-developed	Well-developed	Well-developed	Well-developed	Reduced	Reduced	Reduced	
Ossified epipterygoid	Present	Present	Present	Present	Present	–	Absent	Absent	
Triturating surfaces	Narrow	Narrow	Narrow	Narrow	Narrow	–	Expanded	Expanded	
Exposure of prefrontal on skull roof	Moderate	Moderate	Moderate	Moderate	Moderate	Moderate	Reduced	Absent	
Bony contributions to articular surface of occipital condyle	–	Basioccipital	Basioccipital	Basioccipital	–	–	–	Basioccipital & exoccipitals	
Palatine artery	Absent	Present	–	Absent	Present	–	Absent	Absent	
Pterygoid-basioccipital contact	–	Short	Short	Elongate	Elongate	Elongate	Elongate	Elongate	
Anterior tubera basioccipitalis	Present	Present	Present	Absent	Absent	Absent	Absent	Absent	
Basipterygoid process	Present	Present	Present	Absent	Absent	–	Absent	Absent	

Lakotemys australodakotensis notably shares with Arundelemys dardeni and pleurosternids the following features, which are absent in more advanced baenids (Joyce & Lyson, 2015): an anterior process of the pterygoid, a well-developed external process of the pterygoids, an ossified epipterygoid, relatively narrow triturating surfaces (larger in more advanced baenids), and a moderate exposure of the prefrontal on the skull roof (minor to absent exposure in more advanced baenids). Additional, potentially plesiomorphic similarities between Lakotemys australodakotensis and pleurosternids can be identified but those are not universally present (or absent) among the primary clades of interest or need further investigation to be confirmed. For instance, Lakotemys australodakotensis has a relatively elongate skull reminiscent of those of Pleurosternon bullockii (Evers, Rollot & Joyce, 2020) and Glyptops ornatus (Gaffney, 1979), but differs from the wedge-shaped skull of Uluops uluops (Rollot, Evers & Joyce, 2021a) and more advanced baenids. Similarly, pleurosternids have been suggested to have an articular surface of the occipital condyle formed by the basioccipital, while baenids seem to have a contribution of the exoccipital to that articular surface (Rollot, Evers & Joyce, 2021a). As we are not able to discern the basioccipital from the exoccipitals in Lakotemys australodakotensis, the condition remains unknown in the latter, but in Arundelemys dardeni, although the occipital condyle is broken off the holotype skull (USNM 497740), the extent of the exoccipitals indicate that they likely did not contribute to the articular surface of the occipital condyle (Evers, Rollot & Joyce, 2021). A sulcus palatino-pterygoideus similar to that of pleurodires and posteriorly framed by a well-developed ridge of the parietal is also present in Arundelemys dardeni (Evers, Rollot & Joyce, 2021), Lakotemys australodakotensis, and Pleurosternon bullockii (Evers, Rollot & Joyce, 2020) but the condition in other paracryptodires remains unknown. Lakotemys australodakotensis also shares a rostrum basisphenoidale that is ventrally exposed between the pterygoids, as found in Glyptops ornatus (Gaffney, 1979) and Pleurosternon bullockii (Evers, Rollot & Joyce, 2020). In the latter two taxa, it is unclear if this extensive ventral exposure of the rostrum basisphenoidale prevented the pterygoids from contacting one another for all their length, whereas in Lakotemys australodakotensis, it is likely that the anterior process of the pterygoids actually met along the ventral midline. Finally, Lakotemys australodakotensis clearly shows evidence for the presence of a canal for the palatine artery, a shared feature with Uluops uluops (Rollot, Evers & Joyce, 2021a). The presence vs absence of a palatine artery canal in paracryptodires was recently discussed in detail (see Rollot, Evers & Joyce, 2021a) and the presence of that canal is nevertheless unambiguous in Lakotemys australodakotensis (see Pterygoid above).

Similarities between Arundelemys dardeni, Lakotemys australodakotensis, and advanced baenids are apparent as well, as they share the presence of a posterior process of the pterygoid with an elongate contact with the basioccipital (absent in pleurosternids), the absence of secondary basioccipital tubera formed by the parabasisphenoid along the posterior portion of its suture with the pterygoid (present in pleurosternids), and the absence of a basipterygoid process (present in pleurosternids). Lakotemys australodakotensis also has a deep temporal emargination, similar to that of most baenids, with the exception of Baena arenosa and Neurankylus torrejonensis (unknown for Arundelemys dardeni), but that differs from the low emargination found in pleurosternids (Evans & Kemp, 1975; Evans & Kemp, 1976; Gaffney, 1979; Rollot, Evers & Joyce, 2021a).

Although preliminary, the new observations described above provide some direction for expansion and revision of the character matrix used for inferring relationships within paracryptodires. The impact of this expansion upon our understanding of baenid relationships will be explored in a future contribution. The fact that Arundelemys dardeni and Lakotemys australodakotensis exhibit a combination of features mainly found in pleurosternids and more derived features typical of baenodds potentially place them in a morphologically intermediate position between the two clades. The description of Trinitichelys hiatti as well as an expanded phylogenetic analysis is expected to provide new insights into some paracryptodiran relationships, especially the placement of Arundelemys dardeni, Lakotemys australodakotensis, and Trinitichelys hiatti relative to pleurosternids and advanced baenids.

Conclusions

We provide new insights into the cranial anatomy of the Early Cretaceous (Berriasian to Valanginian) baenid Lakotemys australodakotensis based on µCT scans of referred specimen OMNH 66106. The bony-by-bone segmentation and reconstruction of the skull of Lakotemys australodakotensis allows us to describe its cranial anatomy in detail for the first time and to thoroughly compare it to other known paracryptodiran skulls. The skull of Lakotemys australodakotensis is generally similar to that of Arundelemys dardeni from the Early Cretaceous (Aptian-Albian) of Maryland; these taxa share an intriguing combination of plesiomorphic features, generally characteristic of pleurosternids, and derived features, usually found in more advanced baenids. Lakotemys australodakotensis is also the only baenid for which a canal for the palatine artery can be identified with confidence, suggesting the putative independent loss of that artery at the base of pleurosternids and baenids. The aim of the present study is to provide a better comprehension of basal baenid cranial anatomy, and thus complements the recent descriptions of Arundelemys dardeni (Evers, Rollot & Joyce, 2021) and provides extensive morpho-anatomical details that will be useful for phylogenetic purposes and the investigation of paracryptodiran relationships. The impact of the new anatomical observations made here on paracryptodiran phylogeny will be assessed in a future study along with the relationships of the basal baenids Arundelemys dardeni, Lakotemys australodakotensis, and Trinitichelys hiatti.

We thank the editor Virginia Abdala and the reviewers Donald Brinkman, Ray Chatterji, Marc Jones, and Adan Perez-Garcia for their thorough comments that greatly helped improving the quality of the manuscript. We also thank Matthew Colbert from the University of Texas High-Resolution X-ray Computed Tomography Facility for scanning OMNH 66106.

Institutional Abbreviations

AMNH American Museum of Natural History, New York, New York, USA

CCM Carter County Museum, Ekalaka, Montana, USA

DMNH Denver Museum of Nature and Science, Denver, Colorado, USA

DORCM Dorset County Museum, Dorchester, UK

FMNH Field Museum of Natural History, Chicago, Illinois, USA

IWCMS Dinosaur Isle Museum (formerly Museum of Isle of Wight Geology), Sandown, UK

MCZ Museum of Comparative Zoology, Cambridge, Massachusetts, USA

ND North Dakota Heritage Center, Bismarck, North Dakota, USA

NMMNH New Mexico Museum of Natural History and Science, Albuquerque, New Mexico, USA

OMNH Sam Noble Oklahoma Museum of Natural History, Norman, Oklahoma, USA

TMP Royal Tyrrell Museum of Paleontology, Drumheller, Alberta, Canada

UALVP University of Alberta Laboratory of Vertebrate Paleontology, Edmonton, Alberta, Canada

UCM University of Colorado Museum of Natural History, Boulder, Colorado, USA

UCMP University of California Museum of Paleontology, Berkeley, California, USA

UMMP University of Michigan Museum of Paleontology, Ann Arbor, Michigan, USA

UMNH Natural History Museum of Utah, Salt Lake City, Utah, USA

UMZC University Museum of Zoology, Cambridge, England

USNM United States National Museum of Natural History, Washington, D. C., USA

YPM Yale Peabody Museum of Natural History, New Haven, Connecticut, USA

Additional Information and Declarations

Competing Interests

Author Contributions

Data Availability

The authors declare there are no competing interests.

Yann Rollot conceived and designed the experiments, performed the experiments, analyzed the data, prepared figures and/or tables, authored or reviewed drafts of the paper, and approved the final draft.

Serjoscha W. Evers, Richard L. Cifelli and Walter G. Joyce analyzed the data, authored or reviewed drafts of the paper, and approved the final draft.

The following information was supplied regarding data availability:

The CT data and 3D models are available at MorphoSource: 000379242.

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
