# Peer review of "New insights into the cranial osteology of the Early Cretaceous paracryptodiran turtle Lakotemys australodakotensis"

_PeerJ, doi:10.7717/peerj.13230_

## Round 0.1 · original submission · Minor Revisions

The three reviewers suggested minor revisions, and I concur. Please add in the Methods section a summary of the taxa compared with Lakotemys, and follow the editorial changes suggested by all three reviewers.

Reviewer #3 indicated some problems with the bones' colors and questioned the sculpturing report on the parietal's dorsal surface. Consider adding a close-up photograph of the area. Please also consider adding a paragraph on baenid turtles in general and a phylogenetic tree to clarify some of the taxonomic terminology and phylogenetic hypotheses used.

All reviewer's suggestions are helpful, and following them would improve your study considerably.

·

Basic reporting

no comment

Experimental design

no comment

Validity of the findings

no comment

Additional comments

My only recommendation is that a paragraph be added in the Methods section giving a summary of the taxa that are being compared with Lakotemys. This could include their phylogenetic position within the Paracryptodira and reference to papers in which morphological data is provided.

·

Basic reporting

Dear editor and authors,

In my opinion this is an interesting paper to be published in this Journal, and I only suggest some minor changes:
- Lines 60-63 (i.e. last sentence of the Introduction): I suggest deleting this sentence relative to future studies (or restricting its use to the Conclusions)
- Line 81: The generic name should be italicized
- Line 91: The “finely sculptured skull and shell” should be more detailed described
- Lines 318-323 and others: Although the authors indicate, for some comparisons, taxa with which a character is shared, and others from which the analyzed taxon differs, this is not done for all discussed characters. Please complete all comparisons.
- Line 392: I think that, in the text, it is better not to use abbreviations
- Line 849: Please add “of it holotype” at the end of that sentence.
- Lines 849-854: In both sentences (especially in the first) it is not clear what information is provided for the first time in this paper. I suggest expanding the conclusions or modifying these sentences to clearly indicate what new information is provided.
- Figure 1: Although the specimen was previously published, I suggest adding a photograph in dorsal view and another in ventral view of the skull.

I’m available for any questions that may arise.

Sincerely,

Adán Pérez-García

Experimental design

no comment

Validity of the findings

no comment

Additional comments

no comment

·

Excellent Review

This review has been rated excellent by staff (in the top 15% of reviews)
EDITOR COMMENT
I agree that it is an excellent review. It is detailed and respectful. Several valuable suggestions were made and helped the authors improve their work.

Basic reporting

This manuscript provides a description of the cranial material of a specimen of paracryptodiran turtle Lakotemys australodakotensis from the Early Cretaceous of North America. The description itself is generally well written and clear. The issues we have with the text are minor and easily rectified. The figures are clear, informative, well labelled, and reasonably well integrated with the text. However, the colours chosen for the bones could perhaps be more in contrast with each other: they were difficult to distinguish in a non-colour print out. Also some the scale bars seem wrong (e.g. Figure 5, 7, and 8). The only part of the physical description which we found questionable was the report of sculpturing on the dorsal surface of the parietal. As it is stated in the paper, the skull is deformed and the surface has been eroded, and these structures are only present on one side. A close up photograph of the area (ideally a stereopair) might help provide more confidence that these structures are not the result of postmortem damage. There isn’t a phylogenetic analysis but some characters and their distribution are discussed. The character distribution is used to make some inferences regarding the phylogenetic relationships of Lakotemys australodakotensis with respect to other baeniid turtles and these are well argued. The paper clearly adds important knowledge for baeniid turtles and their evolution. Overall we recommend publication with minor revisions.

# Clear and unambiguous, professional English used throughout?
The article is written in English and generally uses clear, unambiguous, and technically correct text. The writing is generally good but there are several sentences that could perhaps be reduced in length, particularly in the introduction and discussion. There are also some paragraphs in the description that could be broken into two or even three paragraphs (e.g. the section on the parietal).

# Literature references, sufficient field background/context provided?
The introduction and background could certainly be expanded upon to demonstrate how the work fits into the broader field of knowledge. At present the introduction is rather brief and focuses on the species Lakotemys australodakotensis. To make the paper more accessible to wider audience the authors should consider adding a paragraph or at least a few sentences on baenid turtles in general, e.g. a group of extinct testudines known from the Cretaceous and Palaogene of North America considered to be aquatic and comprising over a dozen species. Another useful addition to the introduction would be a phylogenetic tree (as inferred by previous work) as it would clarify some of the taxonomic terminology and phylogenetic hypotheses being referred to. Moreover, given how unstable some taxonomic terms can be, it may be helpful to researchers reading this description in 10-20 years’ time for understanding the context in which it was executed. A table listing the main taxa, their age, and geographic origin, anatomy known, and primary reference would also be a useful addition for readers unfamiliar with the group. In some places it might also be useful for the authors to refer to rather than repeatedly listing individual species.

# Professional article structure, figures, tables. Raw data shared?
The structure of the manuscript generally makes sense. All figures are relevant to the content of the article, are of sufficient resolution, and are appropriately described and labelled. There are some parts of the results that would fit better in the discussion, such as speculation on the phylogenetic position of Lakotemys australodakotensis. Appropriate raw data and 3D surface files have been made available on Morphosource. The description might be improved by moving all of the comparisons with other turtles to a separate section after a straight description of Lakotemys australodakotensis. However, we appreciate that altering this structure is a non-trivial task and arguably a personal preference.

# Self-contained with relevant results to hypotheses.
Some additional introduction would move the manuscript closer to being ‘self-contained,’ and some tabulation of key character observations and measurements would move the manuscript closer to representing an appropriate ‘unit of publication’. This manuscript could also make it clearer how much of the skull was previously described in Joyce, Rollot, and Cifelli (2020). That would help highlight how much of the information provided here is new.

.

Experimental design

# Original primary research within Aims and Scope of the journal (https://peerj.com/about/aims-and-scope/)
Yes, the manuscript represents original research in the field of palaeontology which is applicable to Biological Science.

# Research question well defined, relevant & meaningful. It is stated how research fills an identified knowledge gap.
The manuscript has a clearly defined research question: what additional anatomical knowledge of Lakotemys australodakotensis is provided by microCT scanning and what does the additional data indicate regarding the evolution of baenid turtles. However, one might argue that the study could go further by adding a more detailed phylogenetic analysis.

# Rigorous investigation performed to a high technical & ethical standard.
The investigation appears to have been carried out appropriately. The material has been examined using microCT and the authors are honest when the CT data is unclear, which is good practice. There are places where the authors could clarify exactly how certain bones articulate with one another in three dimensions (e.g. Jones et al. 2011) and whether their descriptive text is from a ventral or lateral view etc. The scalebars need to be rechecked.
Jones, M.E., Curtis, N., Fagan, M.J., O’Higgins, P. and Evans, S.E., 2011. Hard tissue anatomy of the cranial joints in Sphenodon (Rhynchocephalia): sutures, kinesis, and skull mechanics. Palaeontologia Electronica, 14(2), p.17A.

# Methods described with sufficient detail & information to replicate.
The methods used are generally described with sufficient information to be reproducible by another investigator. However, the authors should clarify what they mean by “interpolated slice-by-slice segmentation”? The manuscript could also be more explicit when it comes to comparative material and species: which specimens and species were examined first hand vs how much of the comparisons were restricted to the published literature? The authors are not obliged to examine every fossil in person, however it is helpful for them to report which they did and which they did not.

.

Validity of the findings

# Impact and novelty not assessed. Meaningful replication encouraged where rationale & benefit to literature is clearly stated.
This manuscript arguably represents an “add-on” to a previous description (Joyce, Rollot, and Cifelli, 2020) which did not include a detailed examination of the CT data. This new manuscript certainly does add a significant amount of new data. However, the text could perhaps make it clearer by providing extra detail regarding what was previously described and what is novel here. The authors could also count how many additional phylogenetic characters can be scored with the new data which they provide. This exercise would help highlight why this newmanuscript is a worthwhile addition to the literature.
Joyce, W.G., Rollot, Y. and Cifelli, R.L., 2020. A new species of baenid turtle from the Early Cretaceous Lakota Formation of South Dakota. Fossil Record, 23(1), pp.1-13.

# All underlying data have been provided; they are robust, statistically sound, & controlled.
The CT models are archived on Morphosource and are already accessible which is excellent practice.

# Conclusions are well stated, linked to original research question & limited to supporting results. The description is well figured and generally well described. The results and conclusions reached are generally supported by the data. Phylogenetic inferences are based on comparison of a specific characters among a subset of key baenid turtles. These are reasonable considering the available data and are appropriately couched as needing further empirical research. However, it would be very useful for these to be tabulated in additional to the text for reference (cf. Roček, Z., 2008). Alternatively, a detailed phylogenetic analysis could be added to the manuscript highlighting the new information with respect to Joyce, Rollot, and Cifelli (2020).
Roček, Z., 2008. The Late Cretaceous frog Gobiates from Central Asia: its evolutionary status and possible phylogenetic relationships. Cretaceous Research, 29(4), pp.577-591.

.

Additional comments

Detailed comments are itemised and listed by abstract, line number, or figure number below:

Abstract
Please insert USA after “South Dakota”

Abstract
We’d prefer “least nested” to early diverging and basal

Line 32
“Joyce, Rollot & Cifelli, 2020” should be in brackets

Line 38
Although would be better than “while” here

Line 39
Presumably you mean “X-ray” micro-computed tomography, e.g. rather than neutron micro-computed tomography.
Jones, M.E., Lucas, P.W., Tucker, A.S., Watson, A.P., Sertich, J.J., Foster, J.R., Williams, R., Garbe, U., Bevitt, J.J. and Salvemini, F., 2018. Neutron scanning reveals unexpected complexity in the enamel thickness of an herbivorous Jurassic reptile. Journal of The Royal Society Interface, 15(143), p.20180039.

Line 40
Please clarify whether you mean external suture seams, internal suture interfaces, or both.

Line 40
Changing “initially rendered difficult because of” to “obstructed” would make this sentence less wordy and add focus to the main point of the sentences: the reason description was difficult.

Line 40
Perhaps you should state why metal oxides are a problem in this context. Was resolution of the scan an issue? What type of rock is the skull preserved in?

Line 42
The word “restricting” might be a better word than “leading” here.

Line 43
Please add a couple of sentences to expand upon “only a brief description of OMNH 66106.” How much of the external anatomy was already described?

Line 46-49
A table could be helpful here to summarise the core species of the group, where they are from, their geological age, what anatomical parts are known of them, and the key references describing them.

Line 55
“As part of a” should be the start of a new paragraph

Line 61
You could simply use “here” instead of “herein”.

Line 70
Should contra be in italics?

Line 72
Amira version what?

Line 72
Please clarify what you mean by “interpolated slice-by-slice segmentation”? Did you segment every slice in one plane or did you segment every other slice and interpolate between them using the Amira interpolation tool? Did you check information in the other two planes? Please provide this information.

Line 87
Please add “USA” after “South Dakota” given that PeerJ is an international journal and (believe it or not) not everyone outside North America knows where South Dakota is.

Line 92
Please consider changing “branches into the skull” to “branches into the ventral surface of the skull”.

Line 94
Perhaps add more information to clarify “well-developed axillary and inguinal buttresses”. Perhaps add “of the carapace”. Also what you mean by “well-developed”. How do they compare to other elements of the carapace?

Line 101
Is “a slightly broader skull” a strong enough character to be diagnostic? Both individual variation as well as taphonomic distortion would make this difficult to use as a diagnostic character. The other diagnostic characters of Lakotemys australodakotensis stated are strong enough on their own. Perhaps it would help to provide specific measurements.

Line 110
You might consider adding a short section here describing what key comparative specimens you have examined first hand and which you compare using only the literature.

Line 112
Any lithology for the matrix? Sandstone? Siltstone?

Lines 121 and 136
As stated in the general comments, given the reported erosion and damage to other parts of the skull, it seems uncertain that the observed ridges and striations do represent ornamentation. A close up photo of the area might prove helpful.

Line 180
A close up figure of this nasal maxilla contact would be useful

Line 217
Please change “The frontal is about twice wider posteriorly than anteriorly” to “The frontal is twice as wide posteriorly as it is anteriorly”

Line 209
Citations to Figure 3 and 4 appear out of order?

Line 248-249
You could provide actual measurements here.

Line 248-306
It seems odd that there is no mention of the ornamentation here.

Line 261
Consider adding a paragraph break after “foramen stapedio-temporale in dorsal view (Fig. 2A).”

Line 262
“Whereas” would be better than “while” given that “while” introduces a time clause.

Line 262
Please consider replacing “suture” with “external suture seam” and clarify whether you mean in lateral view.
Jones, M.E., Curtis, N., Fagan, M.J., O’Higgins, P. and Evans, S.E., 2011. Hard tissue anatomy of the cranial joints in Sphenodon (Rhynchocephalia): sutures, kinesis, and skull mechanics. Palaeontologia Electronica, 14(2), p.17A.

Line 266
Consider changing “to be closer to the original state” to “to more accurately reflect the anatomical arrangement in life”.

Line 270
“rider” should be defined for those unfamiliar with such terminology.

Line 272
Consider adding a paragraph break after “but different from the elongated "rider" one of Eubaena cephalica (Werneburg, Evers & Ferreira, 2021).”

Line 288
Please add “(for cranial nerve V)” after the first time you use the phrase “trigeminal foramen”

Line 295
Excellent example of how external suture seams can be highly misleading. See Jones et al. 2011 for other examples.

Line 301
Please re-read and clarify the text here. Presumably you simply mean that the difference in reconstruction between the right and left sides is because the left-side is less well preserved and this issue led to segmentation uncertainties.

Line 308
Please add “are” between “preserved in OMNH 66106 but” and “damaged”.

Line 323
Consider changing “No contact between the ventral process of the postorbital and pterygoid is apparent in OMNH66106, being prevented by the medial process of the jugal.” To “No contact between the ventral process of the postorbital and pterygoid is apparent in OMNH66106 as it is prevented by the medial process of the jugal.”

Line 319
If you think this short contact with the maxilla is important perhaps consider adding the specific figure or page number after the year of the reference.

Line 326
Presumably you mean in dorsal view rather than “on the skull roof”.

Line 328
Which “ventral process”?

Line 329
Posterior is an adjective so you need to add end: “posterior end”

Line 331
You could collate this information with a summary table of taxa and characters (even if you don’t want to perform a phylogenetic analysis at this time), e.g.
Roček, Z., 2008. The Late Cretaceous frog Gobiates from Central Asia: its evolutionary status and possible phylogenetic relationships. Cretaceous Research, 29(4), pp.577-591.

Line 340
For directness please change “Variation in our reconstructions of the jugal and maxilla is apparent between the two sides of the skull” to “Our reconstructions of the jugal and maxilla differ between the right and left sides””

Line 341
For clarity please change “skull in that this exposure is more extensive in the right jugal.” To “skull in that exposure of the right jugal is greater than exposure of the left jugal”

Line 344
It looks to me like the posterodorsal corner of the maxilla is broken and missing. There is even a shadow on the jugal which might represent the edge of the facet for the maxilla. Can this area be inspected more closely?

Line 348
Please clarify what the shape of the anterior end of the jugal is if known. Is the surface model provided via morphosource complete?
https://www.morphosource.org/concern/parent/000379250/media/000379268

Line 378
You don’t need the qualifier “quite” here. Moreover, this qualifier means different things to different English speakers: in the UK it tends to mean “a lot” whereas in USA it tends to mean “only a little”.
Levshina, N., 2014. Geographic variation of quite+ ADJ in twenty national varieties of English: A pilot study. Yearbook of the German Cognitive Linguistics Association, 2(1), pp.109-126.

Line 398
Providing a subheading for the premaxilla and explicitly stating that it is not preserved is good practice.

Line 415
Please consider replacing “suture” with “external suture seam” and clarify whether you mean in lateral view.
Jones, M.E., Curtis, N., Fagan, M.J., O’Higgins, P. and Evans, S.E., 2011. Hard tissue anatomy of the cranial joints in Sphenodon (Rhynchocephalia): sutures, kinesis, and skull mechanics. Palaeontologia Electronica, 14(2), p.17A.

Line 455
“did not contact” might be better than “were not contacting”

Line 462
The description might be improved by moving all of these comparisons to a separate section after a straight description. However, we appreciate that altering this structure is a non-trivial task and arguably a personal preference.

Line 477
Please clarify what you mean by sutured. Do you mean interdigitated? if so, in what plane?
Please see Jones et al. 2011.

Line 479
Please clarify what the nature of this contact is? Is it a simple butt joint or is there some overlap or interdigitation.

Line 494
Please change “this” to “this feature” or “this trait” (or similar)

Line 515
Consider changing:
“Due to unclear sutures between the basioccipital and exoccipitals, it is not clear if the pterygoid also contacted the exoccipital in this region.”
to
“It is not clear if the pterygoid also contacted the exoccipital in this region because the sutures between the basioccipital and exoccipitals are unclear.”

Line 525
Is there a reference you can cite to support the statement “the pterygoid forms the sulcus cavernosus, which served as the passage for the lateral head vein posteriorly through the braincase”

Line 534
Please consider changing “anteriormost” to “anteriormost foramina”.

Line 539
Is there a reference that you can cite to support the statement:
”consistent with the expectations for a palatine artery canal, thus allowing us to identify it as the canalis caroticus lateralis.”

Line 571
The authors shouldn’t really speculate about phylogenetic position and its implications in the results. Therefore, this section of text should probably be moved to the discussion.

Line 584
For clarity, please qualify what you mean by “this”: “this what?”

Line 625
Please change “this is different” to “this arrangement is different”

Line 784
You should begin the discussion with a few sentences by clarifying what this new description adds to the previous description by Joyce, Rollot, and Cifelli (2020).

Line 786
minor but the entire list of species name is probably not necessary here

Line 851
Plesimorphic might be better than primitive.

Line 833
Starting at “although” should be a new paragraph.

Line 833
It is probably not necessary to relist all of the shared characters as the entire previous part of the discussion covers this in great detail.

Line 848-857
You should add a couple of sentences highlighting the new observations that were not previously reported by Joyce, Rollot, and Cifelli, (2020).

Line 854
Please add a sentence explicitly stating why the presence of “a canal for the palatine artery” is important, e.g. “The presence of the canal provides a potential synapomorphy with Uluops uluops which is supposedly non-beanid paracryptodire turtle (Rollot, Evers & Joyce, 2021a)”.

Line 860
Many thanks for your pre-emptive thank you!

Line 860
Are there any curators or CT technicians that also need to be thanked?

Line 864
Given that PeerJ is an international journal please spell out Oklahoma and add USA.

Line 865
Again, given that PeerJ is an international journal please add USA.

Line 868
Adding the models to Morphosource is excellent practice and we were able to access the scans.

Line 942
The volume number should be in bold.

Figure 1
This figure is useful but would be improved if you can make the matrix a different colour such as brown or orange in Fig 1 A and C? photographs of the specimen or occipital view would also be useful additions.

Also “2cm” should be “2 cm”.
Please change throughout.

Figure 2
“1cm” should be “1 cm”.

Figure 3
If these structures are not covered by matrix a photo of the actual surface should be provided as well.
Also “1cm” should be “1 cm”.
Also “2cm” should be “2 cm”.

Figure 4
“5mm” should be “5 mm”.
“3mm” should be “3 mm”.

Figure 5
Please recheck the scale bars. The one for A is described as 5 mm but seems to include 8 squares whereas the on ein B is described as 1 cm but seems to include 6 squares. Something is wrong.
Also please change “5mm” to “5 mm” etc.

Figure 6
Excellent! We love it. However, the foramen should be labelled with CNV if you are indeed confident that it is the trigeminal nerve.
“2cm” should be “2 cm”.

Figure 7
Please recheck the scale bars. The one for A is labelled as 5 mm but seems to include 4 squares whereas the on ein B is labelled as 5 mm but seems to include 6 squares. Something is wrong.
Also please change “5mm” to “5 mm” etc.

Figure 8
Please recheck the scale bars. The scale bar is labelled as 1 cm but seems to include 6 squares.
Also please change “1cm” to “1 cm”.

Tables
There are currently no tables. However, two tables would be a worthwhile addition:
1. a table of important baenid taxa listing their geographic locality information, geologic age, anatomical representation, and primary reference.
2. a table of measurements (e.g. skull width at the quadrates, quadrate height, foramen magnum width, occipital condyle width) would be useful to include.



.

---

## Round 0.2 · Minor Revisions

Although both reviewers suggest acceptance of your manuscript, a few minor points remain - as indicated by our reviewer 3 -by fixing them, you will improve the manuscript, and we will be able to bring this process to a successful conclusion.

·

Basic reporting

The authors have made or justified all minor changes suggested by all reviewers, so that I believe this manuscript can be accepted in its current version.

Experimental design

no comment

Validity of the findings

no comment'

Additional comments

no comment

·

Basic reporting

As previously communicated, this manuscript provides a description of the cranial material of a specimen of paracryptodiran turtle Lakotemys australodakotensis from the Early Cretaceous of North America. The authors have done a great job of responding to our previous comments. The additional text, tables, and phylogenetic tress elevate the manuscript significantly. The issues regarding the scale bars appear to be fixed. The extra text in the introduction and discussion helps provide context to the description. I anticipate that this manuscript will make a very useful and well cited publication.

# 1.1. Clear and unambiguous, professional English used throughout? #
The article is written in English and uses clear, unambiguous, and technically correct text. I still think some of the sentences would be better broken in two but pulling threads at this stage might cause more harm than good and the text is certainly good enough. There are also some paragraphs in the discussion that could be broken into two or even three.

# 1.2. Literature references, sufficient field background/context provided? #
The introduction is only slightly longer than before but it is significantly better at showing how the work fits into the broader field of knowledge. The phylogenetic trees in figure 1 are an extremely useful addition and will help future scientists understand the context in which this description was made. The table of comparative taxa is also an excellent addition.

# 1.3. Professional article structure, figures, tables. Raw data shared? #
The structure of the manuscript is fine.

# 1.4. Self-contained with relevant results to hypotheses? #
The additional information added to the manuscript has elevated it to being ‘self-contained,’ and an appropriate ‘unit of publication’. In fact, this manuscript set a great benchmark for similar descriptive papers to aspire to.

Experimental design

# 2.1 Original primary research within Aims and Scope of the journal (https://peerj.com/about/aims-and-scope/)? #
Yes, the manuscript represents original research in the field of palaeontology which is applicable to Biological Science.

# 2.2. Research question well defined, relevant & meaningful. It is stated how research fills an identified knowledge gap? #
The manuscript has a clearly defined research question: what additional anatomical knowledge of Lakotemys australodakotensis is provided by microCT scanning and what does the additional data indicate regarding the evolution of baenid turtles. One might argue that the study could go further by adding a more detailed phylogenetic analysis but the addition of Table 3 is certainly good enough.

# 2.3 Rigorous investigation performed to a high technical & ethical standard? #
As previously written the investigation appears to have been carried out appropriately. The material has been examined using microCT and the authors are explicit and honest when the CT data is unclear, which is good practice.


# 2.4 Methods described with sufficient detail & information to replicate? #
The authors have now clarified what they meant by “interpolated slice-by-slice segmentation” which is good. The additional table of comparative material and species is excellent.

Validity of the findings

# 3.1 Impact and novelty not assessed. Meaningful replication encouraged where rationale & benefit to literature is clearly stated? #
As previously stated this manuscript arguably represents an “add-on” to a previous description (Joyce, Rollot, and Cifelli, 2020) which did not include a detailed examination of the CT data. However, this new manuscript certainly does add a significant amount of new data and it now included enough background information to make it a really useful piece of work that significantly adds to the previous description.

# 3.2 All underlying data have been provided; they are robust, statistically sound, & controlled? #
As stated in our first review, the CT models are archived on Morphosource and are already accessible which is excellent practice.

# 3.3 Conclusions are well stated, linked to original research question & limited to supporting results ? #

The description is well figured and well described. The results and conclusions reached are reasonable and supported by the data. Phylogenetic inferences are based on comparison of a specific characters among a subset of key baenid turtles. These are reasonable considering the available data and are appropriately couched as needing further empirical research. The comparisons are now presented in an easy to read table which is extremely useful. The authors deseve congratulations.

Additional comments

4.1. General responses to the “Rebuttal letter”
The authors make a very reasonable case for keeping the figure colours as they are. We didn’t intended to push the issue, only raise it. However, we think it is wrong to assume that “virtually all readers” will read the publication on screen. The habits, resources, and opportunities available to people across the world is diverse. Nevertheless, the colour differences are still good enough even in black and white. We admit that our own publications are not beyond criticism.
We are glad that the scale bars could be fixed. We are happy to help.
If photographs wouldn’t provide meaningful images of the ornamentation then there is no need to provide them. Thanks for clarifying.
We are glad you re-checked the manuscript and we accept that it is certainly good enough. However, we do caution that having “two native American English speakers” as co-authors doesn’t necessarily give you a free pass because native speakers can sometime miss issues that cause needless problems for ESL. The re-checks that you’ve done (by both native and non-native speakers) are more important. We do list a couple of further minor suggestions below.
The addition of table 1 is great.
I looks forwards to seeing the future manuscript that includes the phylogenetic analysis. The new figure 1 and table 1 is enough here.
We are gratefully that you clarified what you meant by “interpolated slice-by-slice segmentation” particularly as it differs from what one of us assumed.
The table that lists all comparative material used is excellent!
I enjoyed reading the revised version and look forwards to seeing it published.

4.2 LINE BY LINE COMMENTS
Detailed comments are itemised and listed by abstract, line number, or figure number below according to the new pdf (not the track changes document):

Line 46
I’ve generally been advised not to refer to references as if they are people and not to make references the subject of the sentence. However, maybe it’s OK here.

Line 86
I don’t understand why the authors use “herein” instead of “here”. It’s not as bad as using “utilize” instead of “use” but it still seems like a waste of letters to me for zero gain. Obviously not a big deal though.

Line 809
Perhaps add “Specimen” in front of OMNH 66106 because it’s the beginning of a paragraph.

Line 809
This first paragraphs seems excessively long. Consider whether it is worth breaking in two.

Line 868
Because the sentence is at the start of a new paragraph and not at the end of a paragraph listing the observations I think you should consider changing “these observations” to “the new observations described above” or maybe “the new observations described here”

Line 892
Please cite a publication to support the statement “and thus complements the recent descriptions of Arundelemys dardeni”

---

## Round 0.3 · accepted · Accept

I appreciate your consideration of all suggestions made by our reviewer. I am happy to accept your work.

Please note that in ln. 633 of the tracked changes manuscript some typos persist: possobility that the epipterygoid in baenodds..